# Nonparametric Instrumental Variable Regression through Stochastic Approximate Gradients

**Yuri R. Fonseca**\*
Decision, Risk and Operations
Columbia University
New York, NY
yfonseca23@gsb.columbia.edu

**Caio F. L. Peixoto**\*
School of Applied Mathematics
Getulio Vargas Foundation
Rio de Janeiro, RJ, Brazil
caio.peixoto@fgv.br

**Yuri F. Saporito**\*
School of Applied Mathematics
Getulio Vargas Foundation
Rio de Janeiro, RJ
yuri.saporito@fgv.br

## Abstract

Instrumental variables (IVs) provide a powerful strategy for identifying causal effects in the presence of unobservable confounders. Within the nonparametric setting (NPIV), recent methods have been based on nonlinear generalizations of Two-Stage Least Squares and on minimax formulations derived from moment conditions or duality. In a novel direction, we show how to formulate a functional stochastic gradient descent algorithm to tackle NPIV regression by directly minimizing the populational risk. We provide theoretical support in the form of bounds on the excess risk, and conduct numerical experiments showcasing our method's superior stability and competitive performance relative to current state-of-the-art alternatives. This algorithm enables flexible estimator choices, such as neural networks or kernel based methods, as well as non-quadratic loss functions, which may be suitable for structural equations beyond the setting of continuous outcomes and additive noise. Finally, we demonstrate this flexibility of our framework by presenting how it naturally addresses the important case of binary outcomes, which has received far less attention by recent developments in the NPIV literature.

## 1 Introduction

Causal inference from observational data presents unique challenges, primarily due to the potential for confounding variables that can affect both outcomes and variables of interest. The unconfoundedness assumption, crucial in this context, posits that all confounding variables are observed and properly accounted for, allowing for an unbiased estimation of causal effects. However, in many real-world scenarios, this assumption is difficult to satisfy. When this is the case, approaches that rely on *instrumental variables* (IVs) — quantities that are correlated with the variable of interest (relevance condition), do not affect the outcome in any other way (exclusion condition) and are independent of the unobservable confounders — offer a way to identify causal effects even when unobserved confounders exist. As a concrete example, suppose we want to estimate the impact of years of education on earnings. Most likely, there are unobservable factors, such as omitted ability, affecting both the decision to study and income. In this case, changes in compulsory schooling laws could be used as an instrument [2].

---

\*Alphabetical order.

38th Conference on Neural Information Processing Systems (NeurIPS 2024).

While traditional parametric approaches to IV regression often require assumptions about the relationships between variables that may not hold in practice, *nonparametric* IV (NPIV) models can adapt to the intrinsic structure of the data, allowing for a more nuanced understanding of causal relationships. For this reason, there has been a recent boost of new algorithms applied to the NPIV estimation problem and its theoretical properties. The challenge is that NPIV estimation is an ill-posed inverse problem [28, 8, 9], and recent methods aim to incorporate developments from predictive models, e.g., deep learning and kernel methods, while also accounting for the particular structure of the inverse problem at hand.

In this work, we deviate from previous approaches to NPIV estimation which end up minimizing *empirical* versions of the populational risk [28, 19, 34, 25], or trying to satisfy large collection of empirical moment restrictions [18, 24, 6]. In our formulation, we compute an analytical functional gradient for the actual *populational* risk. After formulating a consistent estimator for this gradient, we apply stochastic gradient descent in a certain function space to recover the effect of $X$ on the outcome $Y$.

The rest of the paper is structured as follows. We conclude Section 1 with a thorough discussion of the previous works on NPIV and our contributions. Next, we provide the basic setting for NPIV regression in Section 2. In Section 3 we detail the associated risk minimization problem and analytically compute its stochastic gradient. Based on these results, we introduce our method in Section 4, with accompanying practical considerations and theoretical support. Numerical experiments are reported on Section 5, where our algorithm is compared to state-of-the-art machine learning methods for NPIV. Finally, in Section 6 we study the case of binary outcomes, showing how our method naturally addresses this scenario under the current assumptions made in the literature. Proofs of all our results are presented in Appendix A, while Appendix B contains formal comparisons with existing methods and Appendix C provides additional implementation details.

## 1.1 Previous work

Many traditional approaches to IV estimation, like the Two-Stage Least Squares (2SLS) method, rely on linear models for treatment estimation and counterfactual prediction functions (see [3, 40] for a thorough survey of classical IV estimation). These approaches, while efficient for estimating policy effects, depend on strong parametric assumptions. Nonparametric extensions of 2SLS then attempt to introduce model flexibility by utilizing linear projections onto known basis or sieve functions (as in [28, 27]) or kernel-based estimators (as in [17, 13]). However, these traditional methods face limitations in large, high-dimensional datasets as they are sensible to the particular choice of sieve functions and number of basis elements (see the discussion in [19]).

In order to propose a scalable method, [19] introduces DeepIV, a generalization of 2SLS which employs neural networks in each step of the two-stage procedure. Although their algorithm is more suitable for high dimensional data, the authors do not provide theoretical guarantees. Deep learning estimators have also appeared within methods focused on the Generalized Method of Moments (GMM), which leverage moment restrictions imposed by the IV to obtain an estimator. In [6], the authors propose DeepGMM, a reformulation of the optimal weighted GMM problem as a minimax problem. They rely on the identification assumption to demonstrate consistency results, provided their algorithm is able to nearly solve a smooth zero-sum game.

In another direction, some methods exploit developments in the RKHS[2] literature and apply them to NPIV estimation. The KIV algorithm in [34] transforms the problem into two-stage kernel ridge regression through a kernel mean embedding of the distribution of the endogenous covariate given the instrument. In [25], the authors take insight from two-stage problems in stochastic programming and propose DualIV, an algorithm which uses Fenchel duality to transform NPIV into a convex-concave saddle point problem, for which a closed-form RKHS algorithm is presented. Inspired by the gradient boosting literature, [4] propose an algorithm that iteratively fits base learners in a boosting fashion, but still needs a post-processing step where the basis functions are held fixed and the weights are optimized.

We point out that most of the cited works have some flexibility issues. While kernel based methods impose a higher computational cost as the number of data points grows and may exhibit poor

---

[2]Reproducing Kernel Hilbert Space

performance in high dimensions [35, 25], Deep Learning based algorithms can suffer from high variance and instability if the amount of available data is *small*, as was seen in [25, 4].

Furthermore, in many applications of interest, ranging from consumer behavior [30] to epidemiology [23], the outcome is binary; for a review, see [12]. In these scenarios, a straightforward application of the quadratic loss function would result in a misspecification of the problem, potentially leading to erroneous estimates. The above NPIV methods strongly leverage the additive structure of the problem formulation, and proper extensions would require significant effort, a detailed discussion is presented in Appendix B. Traditionally, a common approach for binary outcomes is using a *semi-parametric* specification based on control function [20]. This allows for nonparametric estimation of the distribution of the unobservables together with identification of the causal function's *parameters* [1, 32, 14, 11]. To the best of our knowledge, the only work that addresses *fully nonparametric* IV for binary outcomes is [10], which proposes an estimation strategy based on Tikhonov regularization.

## 1.2 Contribution

We propose a novel algorithm for NPIV estimation, SAGD-IV, which works by minimizing the populational risk through stochastic gradient descent in a function space. Under mild assumptions, we provide finite sample bounds on the excess risk of our estimator. Empirically, we demonstrate through numerical experiments that SAGD-IV achieves state-of-the-art performance and better stability.

We have the freedom to employ a variety of supervised learning algorithms to form the estimator of the stochastic gradient, most notably kernel methods and neural networks. This means our estimator could be tailored to specific scenarios where a particular method is more likely to perform well. Furthermore, our algorithm is naturally able to handle non-quadratic loss functions, which allows us to extend both our algorithm and theoretical guarantees to the binary outcomes case.

## 2 Problem setup and notation

Following most of the recent literature, we start by presenting the problem setup under the additive noise assumption to demonstrate our methodological contribution. Then, in Section 6, we analyze the important case of binary outcomes, a prototypical example of when this assumption does not hold.

Let $(\Omega, \mathcal{A}, \mathbb{P})$ be the underlying probability space, and let $X$ be a random vector of covariates taking values in $\mathcal{X} \subseteq \mathbf{R}^{d_X}$. We assume that the *response variable* $Y$ is generated according to

$$Y = h^\star(X) + \varepsilon, \tag{1}$$

where $\varepsilon \in L^2(\Omega, \mathcal{A}, \mathbb{P})$ and satisfies $\mathbb{E}[\varepsilon] = 0$. We denote by $\mathbb{P}_X$ the distribution of the r.v. $X$ and assume that the *structural function* $h^\star$ belongs to $L^2(\mathcal{X}, \mathcal{B}(\mathcal{X}), \mathbb{P}_X)^3$, which we simply denote by $L^2(X)$. This is a Hilbert space with norm and inner product given by $\|h\|_{L^2(X)}^2 = \mathbb{E}[h(X)^2]$ and $\langle h, g \rangle_{L^2(X)} = \mathbb{E}[h(X)g(X)]$. We assume that $\mathbb{E}[\varepsilon \mid X] \neq 0$, that is, some covariates are endogenous. Finally, we assume the existence of a random vector $Z$, taking values in $\mathcal{Z} \subseteq \mathbf{R}^{d_Z}$ and satisfying

1. $\mathbb{E}[\varepsilon \mid Z] = 0$, the exclusion restriction;
2. $X \not\perp\!\!\!\perp Z$, i.e., $Z$ is relevant.

This makes $Z$ a valid instrumental variable. We define $\mathbb{P}_Z$ and $L^2(Z)$ analogously to $\mathbb{P}_X$ and $L^2(X)$. We further consider the mild assumption that $X$ and $Z$ have a joint density denoted by $p_{X,Z}$. Our goal is to estimate $h^\star$ based on i.i.d. samples from the joint distribution of $X, Z$ and $Y$.

As we have listed in the introduction, there are a few different approaches to estimate $h^\star$. We will follow here the original one of [28], in which we take the expected value of Equation (1) conditioned on $Z$ to get

$$\mathbb{E}[Y \mid Z] = \mathbb{E}[h^\star(X) \mid Z]. \tag{2}$$

This motivates us to define the conditional expectation operator $\mathcal{P} : L^2(X) \to L^2(Z)$ given by

$$\mathcal{P}[h](z) = \mathbb{E}[h(X) \mid Z = z].$$

---

[3] We denote by $\mathcal{B}(\mathcal{X})$ the Borel $\sigma$-algebra in $\mathcal{X}$.

This is a bounded linear operator which satisfies $\|\mathcal{P}\|_{\mathrm{op}} \leq 1$ (the operator norm) and whose adjoint $\mathcal{P}^* : L^2(Z) \to L^2(X)$, that is also a bounded linear operator, is given by $\mathcal{P}^*[g](x) = \mathbb{E}[g(Z) \mid X = x]$. Defining $r(Z) = \mathbb{E}[Y \mid Z]$, we can then rewrite Equation (2) as

$$r = \mathcal{P}[h^\star]. \tag{3}$$

This is a Fredholm integral equation of the first kind [22] and, as such, poses an ill-posed linear inverse problem.

In this context, a common assumption [13, 7, 34] made about $\mathcal{P}$ is compactness. We also need it here, noting that compact operators with infinite dimensional range provide prototypical examples of ill-posed inverse problems [8]. However, we wish to phrase this assumption in a different, albeit equivalent [8], form:

**Assumption 2.1.** Let

$$\Phi(x, z) = \frac{p_{X,Z}(x, z)}{p_X(x)p_Z(z)} \tag{4}$$

denote the ratio of joint over product of marginals densities for the $X$ and $Z$ variables, with the convention that $0/0 = 0$. We assume that this kernel has finite $L^2(\mathbb{P}_X \otimes \mathbb{P}_Z)$ norm, that is,

$$\|\Phi\|_{L^2(\mathbb{P}_X \otimes \mathbb{P}_Z)}^2 = \int_{\mathcal{X} \times \mathcal{Z}} \Phi(x, z)^2 p_X(x) p_Z(z) \, \mathrm{d}x \mathrm{d}z < \infty.$$

This is equivalent to assuming that the conditional expectation operator $\mathcal{P} : L^2(X) \to L^2(Z)$ is Hilbert-Schmidt and, hence, compact.

## 3 The risk and its gradient

Motivated by Equation (3), we introduce a pointwise loss function $\ell : \mathbf{R} \times \mathbf{R} \to \mathbf{R}$ and define the associated *populational risk measure*[4] $\mathcal{R} : L^2(X) \to \mathbf{R}$ as

$$\mathcal{R}(h) = \mathbb{E}[\ell(r(Z), \mathcal{P}[h](Z))]. \tag{5}$$

The example the reader should keep in mind is the squared loss function $\ell(y, y') = \frac{1}{2}(y - y')^2$, although, as we will see, other examples could be used depending on the particular regression setting. Our goal is to solve the NPIV regression problem by solving

$$\inf_{h \in \mathcal{H}} \mathcal{R}(h),$$

where $\mathcal{H}$ is a closed, convex, bounded subset of $L^2(X)$ such that $h^\star \in \mathcal{H}$. We also require $0 \in \mathcal{H}$. For future reference, we state these conditions:

**Assumption 3.1** (Regularity of $\mathcal{H}$)**.** The set $\mathcal{H}$ is a closed, convex, bounded subset of $L^2(X)$, which contains the origin and satisfies $h^\star \in \mathcal{H}$.

The only part of this assumption which concerns the data generating process is $h^\star \in \mathcal{H}$, which essentially means that the set $\mathcal{H}$ is large enough.[5]

For $\mathcal{H}$ satisfying Assumption 3.1, we let $D \triangleq \operatorname{diam} \mathcal{H} < \infty$, so that $\|h\| < D$ for every $h \in \mathcal{H}$. One possible choice for the set $\mathcal{H}$ is the $L^\infty(X)$ ball contained in $L^2(X)$, that is

$$\mathcal{H} = \left\{ h \in L^2(X) : \|h\|_\infty \leq A \right\}, \tag{6}$$

where $A > 0$ is a constant. This set is obviously convex and bounded in the $L^2(X)$ norm. It can be shown that it is also closed, but not necessarily compact, a restriction on the search set imposed in [28]. This can be seen by taking a $\|\cdot\|_\infty$–bounded orthonormal basis for $L^2(X)$, if one exists. We denote by $\operatorname{proj}_{\mathcal{H}}$ the orthogonal projection onto $\mathcal{H}$. In case $\mathcal{H}$ is given by Equation (6), we have the explicit formula[6] $\operatorname{proj}_{\mathcal{H}}[h] = (h^+ \wedge A) - (h^- \wedge A)$.

We now state all the assumptions needed on the pointwise loss $\ell$. We denote by $\partial_2$ a partial derivative with respect to the second argument.

---

[4]We note that a more fitting name for this object would be *projected risk measure*, since we are projecting $h$ onto the instrument space before applying the loss function. Concerning this matter, we chose to follow the terminology used in the current NPIV literature, which refers to this projected risk simply as "risk".

[5]This part could be generalized to $(h^\star + \ker \mathcal{P}) \cap \mathcal{H} \neq \emptyset$, so that there exists $h \in \mathcal{H}$ such that $\mathcal{R}(h) = \mathcal{R}(h^\star)$, without prejudice to any of the theoretical results.

[6]Here we use the notation $h^+ = \max\{h, 0\}, h^- = (-h)^+$ and $a \wedge b = \min\{a, b\}$.

**Assumption 3.2** (Regularity of $\ell$)**.**

1. The function $\ell : \mathbf{R} \times \mathbf{R} \to \mathbf{R}$ is convex and $C^2$ with respect to its second argument;

2. The function $\ell$ has Lipschitz first derivative with respect to the second argument, i.e., there exists $L \geq 0$ such that, for all $y, y', u, u' \in \mathbf{R}$ we have

$$|\partial_2\ell(y, y') - \partial_2\ell(u, u')| \leq L(|y - u| + |y' - u'|).$$

Some useful facts which follow immediately from these assumptions are presented in Appendix A.

## 3.1 Gradient computation

Here we divert from the previous literature and proceed to compute a functional stochastic gradient for the risk $\mathcal{R}$. A similar idea has been considered by [15] in the context of statistical inverse problems, however, their setup assumes that the operator posing the inverse problem is known, which greatly simplifies the analysis and cannot be directly applied to the NPIV problem. We start by providing an analytical formula for $\nabla \mathcal{R}(h)$:

**Proposition 3.3.** *The risk $\mathcal{R}$ is Fréchet differentiable and it's gradient satisfies*

$$\nabla\mathcal{R}(h) = \mathcal{P}^*[\partial_2\ell(r(\cdot), \mathcal{P}[h](\cdot))] \in L^2(X), \tag{7}$$

*where $\mathcal{P}^* : L^2(Z) \to L^2(X)$ is the adjoint of the operator $\mathcal{P}$.*

Both $\mathcal{P}$ and $\mathcal{P}^*$ are unknown operators which commonly appear in NPIV estimation [8, 13], with Nadaraya-Watson kernels [26, 39] being a classical option for estimating them. However, the fact that they are nested in Equation (7) is undesired, and may impose extra computational costs, especially considering that $\partial_2\ell$ is nonlinear in the second argument for non-quadratic losses. We overcome this difficulty by leveraging the following characterization of the gradient:

**Corollary 3.4.** *The gradient of the populational risk satisfies*

$$\nabla\mathcal{R}(h)(x) = \mathbb{E}[\Phi(x, Z)\partial_2\ell(r(Z), \mathcal{P}[h](Z))], \tag{8}$$

*where $\Phi$ is defined as in Equation* (4).

The benefit of our approach is that Equation (8) enables a more computationally efficient way to estimate the gradient of the risk functional, as we explain next. From Equation (8), for a given $x \in \mathcal{X}$, the random variable $\Phi(x, Z)\partial_2\ell(r(Z), \mathcal{P}[h](Z))$ is an unbiased stochastic estimator of $\nabla\mathcal{R}(h)(x)$. Our stochastic *approximate* gradient is then built using estimators $\widehat{\Phi}, \widehat{r}$ and $\widehat{\mathcal{P}}$ of $\Phi, r$ and $\mathcal{P}$ respectively. With this notation, given a sample $Z$, we consider

$$\widehat{\nabla\mathcal{R}(h)}(x) = \widehat{\Phi}(x, Z)\partial_2\ell(\widehat{r}(Z), \widehat{\mathcal{P}}[h](Z)). \tag{9}$$

We have then substituted the estimation of the operator $\mathcal{P}^*$ by the simpler problem of estimating the ratio of densities $\Phi$ and computing its product with $\partial_2\ell(\widehat{r}(Z), \widehat{\mathcal{P}}[h](Z))$. Moreover, density ratio estimation is an area of active research within the machine learning community, which makes developments in this direction immediately translatable into benefits to our method.

## 4 Algorithm: implementation and theory

In Algorithm 1 we present *Stochastic Approximate Gradient Descent IV (SAGD–IV)*[7], a method for estimating $h^\star$ using the approximation given by Equation (9).

---

[7]The nomenclature Stochastic Approximate Gradient Descent (SAGD) first appeared in another paper addressing an unrelated topic, [31], where Langevin dynamics are employed to obtain approximate gradients when samples of the underlying random variable are not available.

**Algorithm 1** SAGD–IV

---

**Input:** Samples $\left\{(\boldsymbol{z}_m)_{m=1}^M\right\}$. Estimators $\widehat{\Phi}, \widehat{r}$ and $\widehat{\mathcal{P}}$. Sequence of learning rates $(\alpha_m)_{m=1}^M$. Initial guess $\widehat{h}_0 \in \mathcal{H}$.
**Output:** $\widehat{h}$
**for** $1 \le m \le M$ **do**
    Set $u_m = \widehat{\Phi}(\cdot, \boldsymbol{z}_m)\partial_2\ell\left(\widehat{r}(\boldsymbol{z}_m), \widehat{\mathcal{P}}[\widehat{h}_{m-1}](\boldsymbol{z}_m)\right)$
    Set $\widehat{h}_m = \mathrm{proj}_{\mathcal{H}}\left[\widehat{h}_{m-1} - \alpha_m u_m\right]$
**end for**
Set $\widehat{h} = \frac{1}{M}\sum_{m=1}^M \widehat{h}_m$

---

*Remark* 4.1. We note the fact that the internal loop of Algorithm 1 only needs samples from the instrumental variable $Z$ to unfold.

For our theoretical analysis, all we require of the estimators of $\Phi$, $r$ and $\mathcal{P}$ is the following:

**Assumption 4.2** (Properties of $\widehat{\Phi}, \widehat{r}$ and $\widehat{\mathcal{P}}$)**.**

1. $\widehat{\Phi}, \widehat{r}$ and $\widehat{\mathcal{P}}$ are computed using a dataset $\mathcal{D}$, composed of $N$ samples of the triplet $(X, Z, Y)$, which are independent from the $Z$ samples used in Algorithm 1.

2. $\widehat{r} \in L^2(Z)$ a.s.;

3. $\widehat{\mathcal{P}} : L^2(X) \to L^2(Z)$ is a bounded linear operator a.s.

4. $\|\widehat{\Phi}\|_\infty \triangleq \underset{(x,z)\in\mathcal{X}\times\mathcal{Z}}{\mathrm{ess\,sup}} |\widehat{\Phi}(x,z)| < \infty$. This implies, in particular, $\|\widehat{\Phi}\|_{L^2(\mathbb{P}_X\otimes\mathbb{P}_Z)} < \infty$.

## 4.1 On computing $\widehat{\mathcal{P}}, \widehat{\Phi}$ and $\widehat{r}$

Algorithm 1 is purposefully agnostic to how the estimators $\widehat{\mathcal{P}}, \widehat{\Phi}$ and $\widehat{r}$ are obtained, in order to provide the user with sufficient modeling flexibility. In what follows, we will explain the options we considered for accomplishing these tasks in the experiments section. A more detailed description is given in Appendix C.

Estimating $\Phi$ is a interesting problem in itself as it is a ratio of densities, and it has drawn significant attention from the Machine Learning community. Here, we consider kernel and neural network methods, regarded as the state of the art. On the other hand, not many options are available to compute $\widehat{\mathcal{P}}$, which means estimating all possible regression of functions of $X$ over $Z$. Here, we chose to use kernel mean embeddings, as was developed [34]. Finally, estimating $r$ is the simplest procedure in our method as it is a regression of $Y$ over $Z$. Similarly to $\Phi$, we also consider kernel and neural network methods.

Based on these options, we decided to specify two variants of SAGD-IV built upon how we compute $\widehat{\Phi}$ and $\widehat{r}$: **Kernel SAGD-IV** uses RKHS methods for both estimators, while **Deep SAGD-IV** employs neural network estimators in these two tasks. The method for computing $\widehat{\mathcal{P}}$ is kernel mean embeddings in both variants.

Under a big data scenario, the dependence of our algorithm on kernel methods to estimate the $\mathcal{P}$ operator could be a limitation. However, we are able to avoid this issue for the estimation of $\Phi$ and $r$. Nonetheless, since our algorithm is agnostic to the specific estimator of $\mathcal{P}$, we could take advantage of any recent developments in the literature of operator-estimation problems.

## 4.2 Risk bound

Since we are directly optimizing the projected populational risk measure, we are able to provide guarantees for $\mathcal{R}(\widehat{h})$ in mean with respect to the training data $\boldsymbol{z}_{1:M} = \{\boldsymbol{z}_1, \ldots, \boldsymbol{z}_m\}$. Our main result is the following:

**Theorem 4.3.** *Let $\widehat{h}_0, \ldots, \widehat{h}_{M-1}$ be generated according to Algorithm 1. Assume that $\ell$ satisfies Assumption 3.2, $\mathcal{H}$ satisfies Assumption 3.1, $\Phi$ satisfies Assumption 2.1 and $\widehat{\Phi}, \widehat{r}, \widehat{\mathcal{P}}$ satisfy Assumption 4.2. Then, if we let $\widehat{h} = \frac{1}{M} \sum_{m=1}^{M} \widehat{h}_{m-1}$, the following bound holds:*

$$\mathbb{E}_{\boldsymbol{z}_{1:M}} \left[ \mathcal{R}(\widehat{h}) - \mathcal{R}(h^\star) \right] \leq \frac{D^2}{2M\alpha_M} + \frac{\xi}{M} \sum_{m=1}^{M} \alpha_m + \tau\sqrt{\zeta}, \tag{10}$$

*where*

$$\xi = \frac{3}{2}\|\widehat{\Phi}\|_\infty^2 \left( C_0^2 + L^2\|\widehat{r}\|_{L^2(Z)}^2 + L^2 D^2 \|\widehat{\mathcal{P}}\|_{\mathrm{op}}^2 \right), \quad \zeta = \|\Phi - \widehat{\Phi}\|_{L^2(\mathbb{P}_X \otimes \mathbb{P}_Z)}^2 + \|r - \widehat{r}\|_{L^2(Z)}^2 + \|\mathcal{P} - \widehat{\mathcal{P}}\|_{\mathrm{op}}^2,$$

$$\tau = 2D \max \left\{ 3(C_0^2 + L^2\mathbb{E}[Y^2] + L^2 D^2), 2L^2\|\widehat{\Phi}\|_\infty^2, 2L^2 D^2 \|\widehat{\Phi}\|_\infty^2 \right\}.$$

It is productive to analyze the RHS of the bound in Equation (10) more carefully. If we choose the sequence $(\alpha_m)$ to satisfy the usual assumptions

$$M\alpha_M \to \infty \quad \text{and} \quad \frac{1}{M} \sum_{m=1}^{M} \alpha_m \to 0$$

as $M \to \infty$, then the first two terms in the sum vanish as $M$ grows. The last term appears due to the fact that we do not know $\Phi, r$ nor $\mathcal{P}$, but it explicitly quantifies how the estimation errors of $\widehat{\Phi}, \widehat{r}$ and $\widehat{\mathcal{P}}$ come together to determine the quality of the final estimator. Furthermore, it converges to zero if the chosen estimation methods are consistent.[8]

*Remark* 4.4 (*$Y$ versus $r(Z)$*). Since $r = \mathcal{P}[h^\star]$, one should compute the risk as in Equation (5), comparing $\mathcal{P}[h](Z)$ with $r(Z)$. However, most works on NPIV [28, 34, 25] compare $\mathcal{P}[h](Z)$ directly with $Y$ instead of $r(Z)$. In Appendix E we discuss this difference and its relationship with the estimator $\widehat{r}$.

*Remark* 4.5 (*Consistency for $h^*$*). Since the NPIV regression is an ill-posed inverse problem, consistency of our algorithm is a very challenging and interesting research question, and it would require additional assumptions. For instance, under strong convexity of the populational risk measure, it is well known that consistency is obtained if the excess risk converges to zero. It can be shown that, for the quadratic loss, strong convexity of the populational risk is equivalent to $\mathcal{P}$ having a continuous inverse.

## 5 Numerical experiments

Here[9], we compare the performance of the two variants of SAGD-IV explained in Section 4.1, Kernel SAGD-IV and Deep SAGD-IV, with that of five other notable algorithms for estimating $h^\star$ in the context of continuous responses: KIV, DualIV, DeepGMM, DeepIV and 2SLS. Hyperparameters and other implementation details are discussed in Appendix C.

### 5.1 Continuous response

To study the performance of our estimator in a continuous response setting, we used the data generating process from [6], which we recall below:

$$Y = h^\star(X) + \epsilon + \delta, \qquad\qquad X = Z_1 + \epsilon + \gamma, \tag{11}$$
$$Z = (Z_1, Z_2) \sim \mathrm{Uniform}([-3, 3]^2), \qquad \epsilon \sim \mathcal{N}(0, 1), \ \gamma, \delta \sim \mathcal{N}(0, 0.1).$$

Thus, the confounding variable is $\epsilon$ and within this specification, four options for $h^\star$ are considered:

$$\textbf{step}: \quad h^\star(x) = \mathbf{1}\{x > 0\}, \qquad\qquad \textbf{abs}: \quad h^\star(x) = |x|,$$
$$\textbf{linear}: \quad h^\star(x) = x, \qquad\qquad\qquad \textbf{sin}: \quad h^\star(x) = \sin(x).$$

---

[8]If one desires to substitute the $\zeta$ term for estimator-specific bounds, rates for $\|\Phi - \widehat{\Phi}\|_{L^2(\mathbb{P}_X \otimes \mathbb{P}_Z)}$ can be found in [36, Theorem 14.16], while bounds for $\|\mathcal{P} - \widehat{\mathcal{P}}\|_{\mathrm{op}}$ are present in [34, Theorem 2], [38, Theorem 5] and [33, Proposition S5]. The term $\|r - \widehat{r}\|_{L^2(Z)}$ is the risk for a simple real-valued regression problem, and various rates are available depending on the chosen regressor and the degree of smoothness assumed on $r$ [16].

[9]Code for the experiments is available at https://github.com/Caioflp/sagd-iv

We adjusted each estimator using 3000 random variable samples (see Remark C.1), and tested them on 1000 samples of $X$. For each method and response function, we evaluated predictions over 20 realizations of the data. Log mean squared error (MSE) box plots and plots of each method's estimator for a randomly chosen realization of the data are displayed in Figure 1.

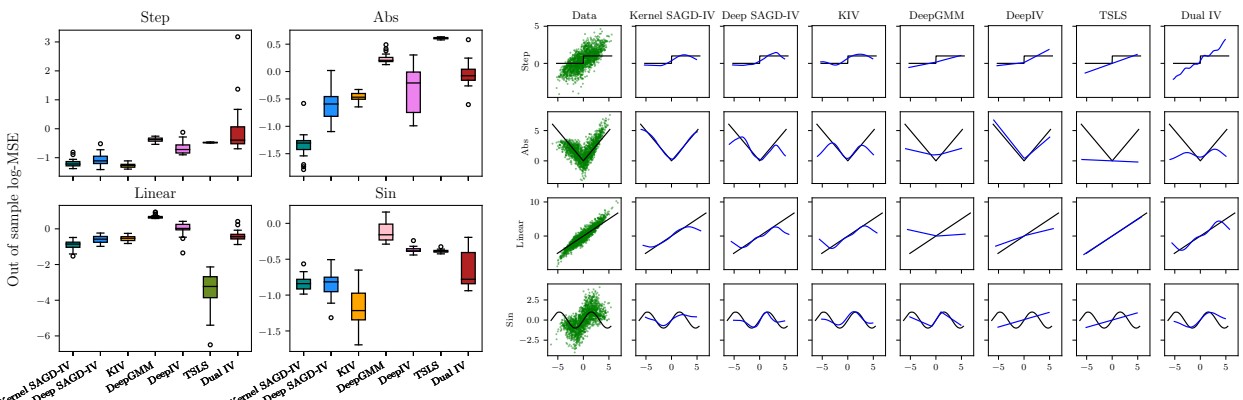

Figure 1: (Left) Log MSE for each model under different response functions. (Right) Plots of each method's estimator in a randomly selected realization of the data. On the left column, we have $Y$ observations in green and the true structural function in black.

The first thing we can notice from Figure 1 is that TSLS dominates the **linear** scenario, as expected, but behaves poorly in all others, showing the importance of nonlinear approaches.

On a second note, both variants of SAGD-IV performed competitively in all scenarios, with the KIV method being a strong competitor in **step** and **sin**. However, its performance was significantly worse for the **abs** response function. Additionally, it is worth noticing the competitive MSEs of the deep learning variant of SADG-IV, which performed consistently better than the other two neural-network-based algorithms.

Furthermore, an important observation is that, within kernel methods, Kernel SAGD-IV had the best stability/performance tradeoff, having the smallest variance across scenarios. In contrast, DualIV exhibited the highest variance. This can be explained by considering DualIV's hyperparameter selection algorithm [25, Section 4]. In order to perform cross-validation to choose the best regularization parameters, the algorithm requires another (different) regularization parameter as an input, which must be chosen without any guidance. The authors state that it was set to a "small constant". This can explain the poor performance, given the importance of regularization parameters for ridge regression algorithms. We note our method does not suffer from this problem.

# 6 Binary Response

Here we outline how our methodology can be applied to a scenario where the quadratic loss is not the natural option: that of binary response models. We show that, by simply modifying the loss $\ell$ in the risk definition and, consequently, in the algorithm, we are able to attack this problem as it is currently formulated in the literature. This is in clear contrast with recent methods for NPIV estimation, which, when applicable, would require significant effort. We discuss the limitations of current methods and possible extensions in Appendix B.

In the binary response model we consider, the only change we need to make to the data generating process is the following:

$$Y = \mathbf{1}\left\{Y^{\star} > 0\right\},$$

where $Y^{\star} = h^{\star}(X) + \varepsilon$. That is, we observe a signal which indicates if the response variable is greater than the threshold zero. Although simple, this type of model captures important applications as, for instance, in economics/consumer behavior, where $Y^{*}$ is the utility the consumer gets from buying a product and we only observe $Y$, the buy/no buy decision. The function $h^{\star}$ captures the

impact of the variable $X$ on the consumers' utility. As usual in the literature, $h^\star$ is interpretable relative to the scale parameter of $\varepsilon$. Here, $X, Z$ and $\varepsilon$ satisfy the same assumptions as before.

In order to use one of the available NPIV regression methods in the literature, we must first identify a suitable risk functional, of the form shown in Equation (5). Define $\eta = h^\star(X) - \mathbb{E}[h^\star(X) \mid Z] + \varepsilon$, so that $\mathbb{E}[\eta \mid Z] = 0$ and
$$Y = \mathbf{1}\left\{\mathbb{E}[h^\star(X) \mid Z] + \eta > 0\right\}.$$
Denoting by $F_{\eta|Z}$ the conditional distribution function of $\eta$ given $Z$, we find
$$\mathbb{E}[Y \mid Z] = \mathbb{P}\left[\mathbb{E}[h^\star(X) \mid Z] + \eta > 0 \mid Z\right] = 1 - F_{\eta|Z}(-\mathbb{E}[h^\star(X) \mid Z]).$$
Assume that the distribution of $\eta$ given $Z$ is always symmetric around zero, so that $1 - F_{\eta|Z}(-y) = F_{\eta|Z}(y)$. We then have $r(Z) = F_{\eta|Z}(\mathcal{P}[h^\star](Z))$. Notice that any notion of risk for a given $h \in L^2(X)$ would have to compare $r(Z)$ and $\mathcal{P}[h](Z)$ through $F_{\eta|Z}$, which is an unknown function. Hence, without making further assumptions, it is not clear how to setup the risk in a useful way, which hinders current NPIV methods from tackling the problem.

As far as we know, *the only assumption in the literature* which allows one to be *fully nonparametric* in $h^\star$ within the binary outcomes setting was first introduced in [10]. It posits that $\eta$ is independent of $Z$, and has known distribution function $F$. Since $\eta = Y^\star - \mathbb{E}[Y^\star \mid Z]$, it is already uncorrelated with any function of $Z$. The assumption extends this to independence, and makes available the resulting distribution function.

Under this assumption, we conclude that $r(Z) = F(\mathcal{P}[h^\star](Z))$ and, by the negative log-likelihood of the Bernoulli distribution, a natural candidate for $\ell$ is
$$\ell(y, y') = \mathrm{BCE}(y, F(y')),$$
where BCE is the binary cross entropy function: $\mathrm{BCE}(p, q) = -[p \log q + (1-p) \log(1-q)]$. Then, the risk becomes
$$\mathcal{R}(h) = \mathbb{E}[\mathrm{BCE}(r(Z), F(\mathcal{P}[h](Z)))]. \tag{12}$$
Whereas it is obvious that the quadratic loss satisfies Assumption 3.2, it is not clear whether the same is true for the loss $\ell(y, y') = \mathrm{BCE}(y, F(y'))$. Nevertheless, if $F(x) = \sigma(x) \triangleq 1/(1 + \exp(-x/\beta))$, the c.d.f. of a logistic distribution with scale parameter $\beta$, one can verify that Assumption 3.2 holds.

## 6.1 Numerical Experiment

We mimicked the continuous response DGP making the necessary modifications:
$$Y = \mathbf{1}\{\mathbb{E}[h^\star(X) \mid Z] + \eta > 0\}, \qquad\qquad X = Z_1 + \eta + \gamma,$$
$$Z = (Z_1, Z_2) \sim \mathrm{Uniform}([-3,3]^2), \qquad \eta \sim \mathrm{Logistic}(0, \beta) \quad \gamma \sim \mathcal{N}(0, 0.1).$$

Here, $\beta$ is a scale parameter which was set to $\sqrt{0.1}$. For response functions, we analyzed the **sin** and **linear** cases, since we must compute $\mathbb{E}[h^\star(X) \mid Z]$ analytically in order to generate the data. This computation is easy for the selected scenarios, and we have
$$\mathbb{E}[X \mid Z = z] = z_1, \quad \mathbb{E}[\sin(X) \mid Z = z] = \frac{\beta\pi \cdot e^{-\frac{0.1}{2}}}{\sinh(\beta\pi)} \sin(z_1)$$

As with the continuous response setting, we present log MSE results and samples from the resulting estimators in Figure 2.

We can immediately see that our kernel based estimator performed remarkably well for both response functions. The binary setup is significantly more challenging, as we are reducing the continuous response $Y$ to a simple binary signal. Kernel SAGD-IV obtained results which are, on average, on par with the continuous response scenario and Deep SAGD-IV had more difficulty uncovering the true structural function.

## 7 Conclusion

In this work, we proposed SAGD-IV, a flexible framework for performing nonparametric instrumental variable regression. It is based on a novel way to tackle the NPIV problem, in which we formulate

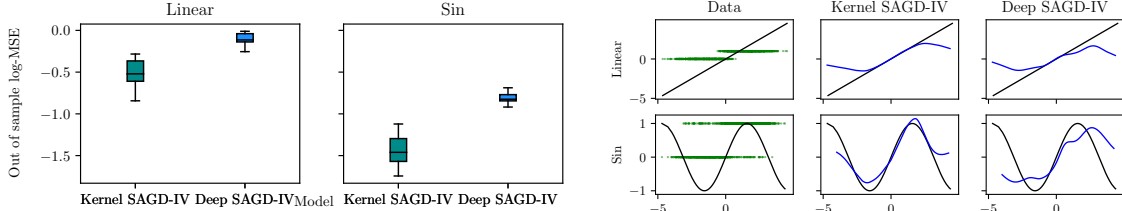

Figure 2: (Left) Log MSE distribution in the binary response DGP. (Right) Plots of each method's estimator in a randomly selected realization of the data for the binary response DGP. On the left column, we have samples from the binary response variable $Y$ in green and the true structural function in black.

the minimization of the populational risk as functional stochastic gradient descent. We are then able to naturally consider non-quadratic loss function and incorporate various regression methods under one formulation. Under mild assumptions, we provided bounds on the excess risk of our estimator. Furthermore, we empirically demonstrated superior stability and state-of-the-art MSEs for continuous outcomes, as well as a promising performance for the binary regression setup.

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

# A  Proofs

**Proposition A.1.** *Assume that $\ell$ satisfies Assumption 3.2. Then:*

1. *Setting $C_0 = |\partial_2 \ell(0,0)|$, we have $|\partial_2 \ell(y,y')| \leq C_0 + L(|y| + |y'|)$, for all $y, y' \in \mathbf{R}$;*

2. *The map $f \mapsto \partial_2 \ell(r(\cdot), f(\cdot))$ from $L^2(Z)$ to $L^2(Z)$ is well-defined and L-Lipschitz.*

3. *The second derivative with respect to the second argument is bounded: $\left|\partial_2^2 \ell(y, y')\right| \leq L$ for all $y, y' \in \mathbf{R}$.*

*Proof.*

1. Write $\partial_2 \ell(y, y') = \partial_2 \ell(y, y') - \partial_2 \ell(0, 0) + \partial_2 \ell(0, 0)$ and apply the triangle inequality as well as Assumption 3.2 Item 2.

2. From the previous item we know this map is well-defined. If $f$ and $g$ belong to $L^2(Z)$, we have

$$\|\partial_2 \ell(r(\cdot), f(\cdot)) - \partial_2 \ell(r(\cdot), g(\cdot))\|_{L^2(Z)}^2 = \mathbb{E}\left[|\partial_2 \ell(r(Z), f(Z)) - \partial_2 \ell(r(Z), g(Z))|^2\right]$$
$$\leq L^2 \mathbb{E}\left[|f(Z) - g(Z)|^2\right]$$
$$= L^2 \|f - g\|_{L^2(Z)}^2.$$

3. It follows directly from Assumption 3.2 Item 2. $\qquad\square$

*Proof of Proposition 3.3.* We start by computing the directional derivative of $\mathcal{R}$ at $h$ in the direction $f$, denoted by $D\mathcal{R}[h](f)$:

$$D\mathcal{R}[h](f) = \lim_{\delta \to 0} \frac{1}{\delta} [\mathcal{R}(h + \delta f) - \mathcal{R}(f)]$$
$$= \lim_{\delta \to 0} \frac{1}{\delta} \mathbb{E}\left[\ell(r(Z), \mathcal{P}[h + \delta f](Z)) - \ell(r(Z), \mathcal{P}[h](Z))\right]$$
$$= \lim_{\delta \to 0} \frac{1}{\delta} \mathbb{E}\left[\ell(r(Z), \mathcal{P}[h](Z) + \delta \mathcal{P}[f](Z)) - \ell(r(Z), \mathcal{P}[h](Z))\right]$$
$$= \lim_{\delta \to 0} \frac{1}{\delta} \mathbb{E}\left[\delta \partial_2 \ell(r(Z), \mathcal{P}[h](Z)) \cdot \mathcal{P}[f](Z)\right.$$
$$\left. + \frac{\delta^2}{2} \partial_2^2 \ell(r(Z), \mathcal{P}[h + \theta f](Z)) \cdot \mathcal{P}[f](Z)^2\right]$$
$$= \mathbb{E}\left[\partial_2 \ell(r(Z), \mathcal{P}[h](Z)) \cdot \mathcal{P}[f](Z)\right]$$
$$+ \lim_{\delta \to 0} \mathbb{E}\left[\frac{\delta}{2} \partial_2^2 \ell(r(Z), \mathcal{P}[h + \theta f](Z)) \cdot \mathcal{P}[f](Z)^2\right]$$
$$= \mathbb{E}\left[\partial_2 \ell(r(Z), \mathcal{P}[h](Z)) \cdot \mathcal{P}[f](Z)\right],$$

where $\theta \in (0, \delta)$ is due to Taylor's formula. The last step is then due to Proposition A.1 Item 3. We can in fact expand the calculation a bit more, as follows:

$$D\mathcal{R}[h](f) = \mathbb{E}\left[\partial_2 \ell(r(Z), \mathcal{P}[h](Z)) \cdot \mathcal{P}[f](Z)\right]$$
$$= \langle \partial_2 \ell(r(\cdot), \mathcal{P}[h](\cdot)), \mathcal{P}[f]\rangle_{L^2(Z)}$$
$$= \langle \mathcal{P}^*[\partial_2 \ell(r(Z), \mathcal{P}[h](\cdot))], f\rangle_{L^2(X)}.$$

This shows that $\mathcal{R}$ is Gâteaux-differentiable, with Gâteaux derivative at $h$ given by

$$D\mathcal{R}[h] = \mathcal{P}^*[\partial_2 \ell(r(\cdot), \mathcal{P}[h](\cdot))].$$

By Proposition A.1 Item 2, we have that $h \mapsto D\mathcal{R}[h]$ is a continuous mapping from $L^2(X)$ to $L^2(X)$, which implies that $\mathcal{R}$ is also Fréchet-differentiable, and both derivatives coincide [29]. Therefore,

$$\nabla \mathcal{R}(h) = \mathcal{P}^*[\partial_2 \ell(r(\cdot), \mathcal{P}[h](\cdot))]. \tag{13}$$

$\qquad\square$

*Proof of Corollary 3.4.* From Proposition 3.3 we know that

$$\nabla\mathcal{R}(h)(x) = \mathcal{P}^*[\partial_2\ell(r(\cdot),\mathcal{P}[h](\cdot))](x) = \mathbb{E}[\partial_2\ell(r(Z),\mathcal{P}[h](Z)) \mid X = x]$$

$$= \int_{\mathcal{Z}} p_{Z|X}(z \mid x)\partial_2\ell(r(z),\mathcal{P}[h](z))\,\mathrm{d}z = \int_{\mathcal{Z}} \frac{p_{X,Z}(x,z)}{p_X(x)}\partial_2\ell(r(z),\mathcal{P}[h](z))\,\mathrm{d}z$$

$$= \int_{\mathcal{Z}} p_Z(z) \cdot \frac{p_{X,Z}(x,z)}{p_X(x)p_Z(z)}\partial_2\ell(r(z),\mathcal{P}[h](z))\,\mathrm{d}z = \mathbb{E}[\Phi(x,Z)\partial_2\ell(r(Z),\mathcal{P}[h](Z))],$$

where we define $\Phi(x,z) = p_{X,Z}(x,z)/(p_X(x)p_Z(z))$ if $p_Z(z) > 0$ and $\Phi(x,z) = 0$ if $p_Z(z) = 0$. Notice we may assume that $p_X(x) > 0$, since the set of $x \in \mathcal{X}$ such that this happens has probability 1, and $\nabla\mathcal{R}(h)$, being an element of $L^2(X)$, is only defined almost surely. □

Before presenting the proof of Theorem 4.3, we need to prove two auxiliary lemmas. To lighten the notation, the symbols $\|\cdot\|$ and $\langle\cdot,\cdot\rangle$, when written without a subscript to specify which space they refer to, will act as the norm and inner product, respectively, of $L^2(X)$.

**Lemma A.2.** *In the procedure of Algorithm 1 we have $u_m \in L^2(X)$ for all $1 \leq m \leq M$ and, furthermore,*

$$\mathbb{E}_{\boldsymbol{z}_{1:M}}[\|u_m\|^2] \leq \rho(\widehat{\Phi},\widehat{r},\widehat{\mathcal{P}}),$$

*where*

$$\rho(\widehat{\Phi},\widehat{r},\widehat{\mathcal{P}}) = 3\|\widehat{\Phi}\|_\infty^2 \left(C_0^2 + L^2\|\widehat{r}\|_{L^2(Z)}^2 + L^2D^2\|\widehat{\mathcal{P}}\|_{\mathrm{op}}^2\right).$$

*Proof.* By Assumption 4.2 we have:

$$\|u_m\|_{L^2(X)}^2 = \|\widehat{\Phi}(\cdot,\boldsymbol{z}_m)\partial_2\ell\left(\widehat{r}(\boldsymbol{z}_m),\widehat{\mathcal{P}}[\widehat{h}_{m-1}](\boldsymbol{z}_m)\right)\|_{L^2(X)}^2$$

$$= \mathbb{E}_X\left[\left|\widehat{\Phi}(X,\boldsymbol{z}_m)\partial_2\ell\left(\widehat{r}(\boldsymbol{z}_m),\widehat{\mathcal{P}}[\widehat{h}_{m-1}](\boldsymbol{z}_m)\right)\right|^2\right]$$

$$\leq \partial_2\ell(\widehat{r}(\boldsymbol{z}_m),\widehat{\mathcal{P}}[\widehat{h}_{m-1}](\boldsymbol{z}_m))^2\|\widehat{\Phi}\|_\infty^2$$

$$< \infty.$$

Hence, $u_m \in L^2(X)$ for all $m$. This computation and Proposition A.1 Item 1 then imply

$$\mathbb{E}_{\boldsymbol{z}_{1:M}}\left[\|u_m\|^2\right] \leq 3\|\widehat{\Phi}\|_\infty^2 \left(C_0^2 + L^2\left(\|\widehat{r}\|_{L^2(Z)}^2 + \|\widehat{\mathcal{P}}[\widehat{h}_{m-1}]\|_{L^2(Z)}^2\right)\right)$$

$$\leq 3\|\widehat{\Phi}\|_\infty^2 \left(C_0^2 + L^2\left(\|\widehat{r}\|_{L^2(Z)}^2 + \|\widehat{\mathcal{P}}\|_{\mathrm{op}}^2\|\widehat{h}_{m-1}\|^2\right)\right)$$

$$\leq 3\|\widehat{\Phi}\|_\infty^2 \left(C_0^2 + L^2\left(\|\widehat{r}\|_{L^2(Z)}^2 + D^2\|\widehat{\mathcal{P}}\|_{\mathrm{op}}^2\right)\right)$$

$$= 3\|\widehat{\Phi}\|_\infty^2 \left(C_0^2 + L^2\|\widehat{r}\|_{L^2(Z)}^2 + L^2D^2\|\widehat{\mathcal{P}}\|_{\mathrm{op}}^2\right) \triangleq \rho(\widehat{\Phi},\widehat{r},\widehat{\mathcal{P}}). \qquad \square$$

**Lemma A.3.** *In the procedure of Algorithm 1 we have*

$$\|\mathbb{E}_{\boldsymbol{z}_m}[\nabla\mathcal{R}(\widehat{h}_{m-1}) - u_m]\| \leq \kappa(\widehat{\Phi})\left(\|\Phi - \widehat{\Phi}\|_{L^2(\mathbb{P}_X\otimes\mathbb{P}_Z)}^2 + \|r - \widehat{r}\|_{L^2(Z)}^2 + \|\mathcal{P} - \widehat{\mathcal{P}}\|_{\mathrm{op}}^2\right)^{\frac{1}{2}},$$

*where*

$$\kappa^2(\widehat{\Phi}) \triangleq 2\max\left\{3(C_0^2 + L^2\mathbb{E}[Y^2] + L^2D^2), 2L^2\|\widehat{\Phi}\|_\infty^2, 2L^2D^2\|\widehat{\Phi}\|_\infty^2\right\}.$$

*Proof.* To ease the notation, we define

$$\Psi_m(Z) \triangleq \partial_2\ell(r(Z),\mathcal{P}[\widehat{h}_{m-1}](Z)),$$

$$\widehat{\Psi}_m(Z) \triangleq \partial_2\ell(\widehat{r}(Z),\widehat{\mathcal{P}}[\widehat{h}_{m-1}](Z)).$$

Let's expand the definition of $\|\cdot\|$:

$$\|\mathbb{E}_{\boldsymbol{z}_m}[\nabla\mathcal{R}(\widehat{h}_{m-1}) - u_m]\| = \mathbb{E}_X\left[\mathbb{E}_{\boldsymbol{z}_m}\left[\nabla\mathcal{R}(\widehat{h}_{m-1})(X) - u_m(X)\right]^2\right]^{\frac{1}{2}}$$

$$= \mathbb{E}_X\left[\left(\nabla\mathcal{R}(\widehat{h}_{m-1})(X) - \mathbb{E}_{\boldsymbol{z}_m}[u_m(X)]\right)^2\right]^{\frac{1}{2}}$$

$$= \mathbb{E}_X\left[\left(\mathbb{E}_Z[\Phi(X,Z)\Psi_m(Z)] - \mathbb{E}_{\boldsymbol{z}_m}[\widehat{\Phi}(X,\boldsymbol{z}_m)\widehat{\Psi}_m(\boldsymbol{z}_m)]\right)^2\right]^{\frac{1}{2}}$$

$$= \mathbb{E}_X\left[\left(\mathbb{E}_Z[\Phi(X,Z)\Psi_m(Z) - \widehat{\Phi}(X,Z)\widehat{\Psi}_m(Z)]\right)^2\right]^{\frac{1}{2}},$$

Now we add and subtract $\widehat{\Phi}(X,Z)\Psi_m(Z)$, so that

$$\mathbb{E}_X\left[\left(\mathbb{E}_Z\left[\Phi(X,Z)\Psi_m(Z) - \widehat{\Phi}(X,Z)\widehat{\Psi}_m(Z)\right]\right)^2\right]^{\frac{1}{2}}$$

$$= \mathbb{E}_X\left[\left(\mathbb{E}_Z\left[\Psi_m(Z)(\Phi(X,Z) - \widehat{\Phi}(X,Z)) + \widehat{\Phi}(X,Z)(\Psi_m(Z) - \widehat{\Psi}_m(Z))\right]\right)^2\right]^{\frac{1}{2}}$$

$$\leq \mathbb{E}_X\left[\left(\|\Psi_m\|_{L^2(Z)}\|\Phi(X,\cdot) - \widehat{\Phi}(X,\cdot)\|_{L^2(Z)} + \|\widehat{\Phi}(X,\cdot)\|_{L^2(Z)}\|\Psi_m - \widehat{\Psi}_m\|_{L^2(Z)}\right)^2\right]^{\frac{1}{2}}$$

$$\leq \sqrt{2}\mathbb{E}_X\left[\|\Psi_m\|_{L^2(Z)}^2\|\Phi(X,\cdot) - \widehat{\Phi}(X,\cdot)\|_{L^2(Z)}^2 + \|\widehat{\Phi}(X,\cdot)\|_{L^2(Z)}^2\|\Psi_m - \widehat{\Psi}_m\|_{L^2(Z)}^2\right]^{\frac{1}{2}}$$

$$= \sqrt{2}\left(\|\Psi_m\|_{L^2(Z)}^2\|\Phi - \widehat{\Phi}\|_{L^2(\mathbb{P}_X\otimes\mathbb{P}_Z)}^2 + \|\widehat{\Phi}\|_{L^2(\mathbb{P}_X\otimes\mathbb{P}_Z)}^2\|\Psi_m - \widehat{\Psi}_m\|_{L^2(Z)}^2\right)^{\frac{1}{2}},$$

where

$$\|\Phi\|_{L^2(\mathbb{P}_X\otimes\mathbb{P}_Z)}^2 = \int_{\mathcal{X}\times\mathcal{Z}} \Phi(x,z)^2 p(x)p(z)\,\mathrm{d}x\mathrm{d}z$$

is the norm with respect to the independent coupling of the distributions of $X$ and $Z$. By Proposition A.1.1 we have

$$\|\Psi_m\|_{L^2(Z)}^2 = \mathbb{E}_Z\left[\partial_2\ell(r(Z),\mathcal{P}[\widehat{h}_{m-1}](Z))^2\right]$$

$$\leq \mathbb{E}_Z\left[\left(C_0 + L\left(|r(Z)| + \left|\mathcal{P}[\widehat{h}_{m-1}](Z)\right|\right)\right)^2\right]$$

$$\leq 3\left(C_0^2 + L^2\|r\|_{L^2(Z)}^2 + L^2\|\mathcal{P}[\widehat{h}_{m-1}]\|_{L^2(Z)}^2\right)$$

$$\leq 3\left(C_0^2 + L^2\mathbb{E}[Y^2] + L^2 D^2\right).$$

It is also clear that, by Assumption 4.2,

$$\|\widehat{\Phi}\|_{L^2(\mathbb{P}_X\otimes\mathbb{P}_Z)}^2 \leq \|\widehat{\Phi}\|_\infty^2.$$

Finally, by Assumption 3.2.2 we also have

$$\|\Psi_m - \widehat{\Psi}_m\|_{L^2(Z)}^2 = \mathbb{E}_Z\left[\left(\partial_2\ell(r(Z),\mathcal{P}[\widehat{h}_{m-1}](Z)) - \partial_2\ell(\widehat{r}(Z),\widehat{\mathcal{P}}[\widehat{h}_{m-1}](Z))\right)^2\right]$$

$$\leq 2L^2\left(\|r - \widehat{r}\|_{L^2(Z)}^2 + \|(\mathcal{P} - \widehat{\mathcal{P}})[\widehat{h}_{m-1}]\|_{L^2(Z)}^2\right)$$

$$\leq 2L^2\left(\|r - \widehat{r}\|_{L^2(Z)}^2 + D^2\|\mathcal{P} - \widehat{\mathcal{P}}\|_{\mathrm{op}}^2\right).$$

To combine all terms, we first define

$$\kappa^2(\widehat{\Phi}) \triangleq 2\max\left\{3(C_0^2 + L^2\mathbb{E}[Y^2] + L^2 D^2), 2L^2\|\widehat{\Phi}\|_\infty^2, 2L^2 D^2\|\widehat{\Phi}\|_\infty^2\right\}.$$

Then, it's easy to see that

$$\|\mathbb{E}_{\boldsymbol{z}_m}[\nabla\mathcal{R}(\widehat{h}_{m-1}) - u_m]\| \leq \kappa(\widehat{\Phi})\left(\|\Phi - \widehat{\Phi}\|_{L^2(\mathbb{P}_X\otimes\mathbb{P}_Z)}^2 + \|r - \widehat{r}\|_{L^2(Z)}^2 + \|\mathcal{P} - \widehat{\mathcal{P}}\|_{\mathrm{op}}^2\right)^{\frac{1}{2}},$$

as we wanted to show. $\qquad\square$

We now have everything needed to show the

*Proof of Theorem 4.3.* We start by checking that $\mathcal{R}$ is convex in $\mathcal{H}$: if $h, g \in \mathcal{H}$ and $\lambda \in [0, 1]$, then

$$
\begin{aligned}
\mathcal{R}(\lambda h + (1 - \lambda)g) &= \mathbb{E}[\ell(r(Z), \mathcal{P}[\lambda h + (1 - \lambda)g](Z))] \\
&= \mathbb{E}[\ell(r(Z), \lambda \mathcal{P}[h](Z) + (1 - \lambda)\mathcal{P}[g](Z))] \\
&\leq \lambda \mathbb{E}[\ell(r(Z), \mathcal{P}[h](Z))] + (1 - \lambda)\mathbb{E}[\ell(r(Z), \mathcal{P}[g](Z))] \\
&= \lambda \mathcal{R}(h) + (1 - \lambda)\mathcal{R}(g).
\end{aligned}
$$

By Assumption 3.1, $h^\star \in \mathcal{H}^{10}$. By the Algorithm 1 procedure, we have

$$
\begin{aligned}
\frac{1}{2}\|\widehat{h}_m - h^\star\|^2 &= \frac{1}{2}\left\|\operatorname{proj}_{\mathcal{H}}\left[\widehat{h}_{m-1} - \alpha_m u_m\right] - h^\star\right\|^2 \\
&\leq \frac{1}{2}\|\widehat{h}_{m-1} - \alpha_m u_m - h^\star\|^2 \\
&= \frac{1}{2}\|\widehat{h}_{m-1} - h^\star\|^2 - \alpha_m\langle u_m, \widehat{h}_{m-1} - h^\star\rangle + \frac{\alpha_m^2}{2}\|u_m\|^2.
\end{aligned}
$$

After adding and subtracting $\alpha_m\langle \nabla\mathcal{R}(\widehat{h}_{m-1}), \widehat{h}_{m-1} - h^\star\rangle$, we are left with

$$
\frac{1}{2}\|\widehat{h}_{m-1} - h^\star\|^2 - \alpha_m\langle u_m - \nabla\mathcal{R}(\widehat{h}_{m-1}), \widehat{h}_{m-1} - h^\star\rangle + \frac{\alpha_m^2}{2}\|u_m\|^2 - \alpha_m\langle \nabla\mathcal{R}(\widehat{h}_{m-1}), \widehat{h}_{m-1} - h^\star\rangle.
$$

Applying the first order convexity inequality on the last term give us, in total,

$$
\begin{aligned}
\frac{1}{2}\|\widehat{h}_m - h^\star\|^2 \leq{}& \frac{1}{2}\|\widehat{h}_{m-1} - h^\star\|^2 - \alpha_m\langle u_m - \nabla\mathcal{R}(\widehat{h}_{m-1}), \widehat{h}_{m-1} - h^\star\rangle \\
&+ \frac{\alpha_m^2}{2}\|u_m\|^2 - \alpha_m(\mathcal{R}(\widehat{h}_{m-1}) - \mathcal{R}(h^\star)).
\end{aligned}
$$

Hence, making this substitution and rearranging terms, we get

$$
\begin{aligned}
\mathcal{R}(\widehat{h}_{m-1}) - \mathcal{R}(h^\star) \leq{}& \frac{1}{2\alpha_m}\left(\|\widehat{h}_{m-1} - h^\star\|^2 - \|\widehat{h}_m - h^\star\|^2\right) \\
&+ \frac{\alpha_m}{2}\|u_m\|^2 - \langle u_m - \nabla\mathcal{R}(\widehat{h}_{m-1}), \widehat{h}_{m-1} - h^\star\rangle.
\end{aligned}
$$

Finally, summing over $1 \leq m \leq M$ leads to

$$
\begin{aligned}
\sum_{n=1}^{M}\left[\mathcal{R}(\widehat{h}_{m-1}) - \mathcal{R}(h^\star)\right] \leq{}& \sum_{m=1}^{M}\frac{1}{2\alpha_m}\left(\|\widehat{h}_{m-1} - h^\star\|^2 - \|\widehat{h}_m - h^\star\|^2\right) \\
&+ \sum_{m=1}^{M}\frac{\alpha_m}{2}\|u_m\|^2 \\
&+ \sum_{m=1}^{M}\langle \nabla\mathcal{R}(\widehat{h}_{m-1}) - u_m, \widehat{h}_{m-1} - h^\star\rangle.
\end{aligned}
\tag{14}
$$

The next step is to take the average of both sides with respect to $z_{1:M}$, taking advantage of the independence between $z_{1:M}$ and $\mathcal{D}$, the data used to compute $\widehat{\Phi}, \widehat{r}$ and $\widehat{\mathcal{P}}$. Each summation in the RHS is then bounded separately.

The first summation admits a deterministic bound. By assumption, the diameter $D$ of $\mathcal{H}$ is finite. Hence

$$
\begin{aligned}
\sum_{m=1}^{M}\frac{1}{2\alpha_m}\left(\|\widehat{h}_{m-1} - h^\star\|^2 - \|\widehat{h}_m - h^\star\|^2\right) ={}& \sum_{m=2}^{M}\left(\frac{1}{2\alpha_m} - \frac{1}{2\alpha_{m-1}}\right)\|\widehat{h}_{m-1} - h^\star\|^2 \\
&+ \frac{1}{2\alpha_1}\|\widehat{h}_0 - h^\star\|^2 - \frac{1}{2\alpha_M}\|\widehat{h}_M - h^\star\|^2 \\
\leq{}& \sum_{m=2}^{M}\left(\frac{1}{2\alpha_m} - \frac{1}{2\alpha_{m-1}}\right)D^2 + \frac{1}{2\alpha_1}D^2 = \frac{D^2}{2\alpha_M}.
\end{aligned}
\tag{15}
$$

---

[10]We want to remind the reader that it is enough to assume that there exists $\bar{h} \in (h^\star + \ker\mathcal{P}) \cap \mathcal{H}$. By the definition of $\bar{h}$, we have $\mathcal{R}(\bar{h}) = \mathcal{R}(h^\star)$.

The second summation can be bounded with the aid of Lemma A.2:

$$\mathbb{E}_{\boldsymbol{z}_{1:M}}\left[\sum_{m=1}^{M}\frac{\alpha_m}{2}\|u_m\|^2\right] = \frac{1}{2}\mathbb{E}_{\boldsymbol{z}_{1:M}}\left[\|u_m\|^2\right]\sum_{m=1}^{M}\alpha_m \leq \frac{1}{2}\rho(\widehat{\Phi},\widehat{r},\widehat{\mathcal{P}})\sum_{m=1}^{M}\alpha_m. \qquad (16)$$

Finally, the third summation can be bounded using Lemma A.3. Let $\mathbb{E}_{\boldsymbol{z}_{-m}}$ denote the expectation with respect to $\boldsymbol{z}_1,\ldots,\boldsymbol{z}_{m-1},\boldsymbol{z}_{m+1},\ldots,\boldsymbol{z}_M$ and notice that

$$\mathbb{E}_{\boldsymbol{z}_{1:M}}\left[\langle\nabla\mathcal{R}(\widehat{h}_{m-1})-u_m,\widehat{h}_{m-1}-h^\star\rangle\right] = \mathbb{E}_{\boldsymbol{z}_{-m}}\left[\mathbb{E}_{\boldsymbol{z}_m}\left[\langle\nabla\mathcal{R}(\widehat{h}_{m-1})-u_m,\widehat{h}_{m-1}-h^\star\rangle\right]\right]$$

$$= \mathbb{E}_{\boldsymbol{z}_{-m}}\left[\langle\mathbb{E}_{\boldsymbol{z}_m}\left[\nabla\mathcal{R}(\widehat{h}_{m-1})-u_m\right],\widehat{h}_{m-1}-h^\star\rangle\right]$$

$$\leq \mathbb{E}_{\boldsymbol{z}_{-m}}\left[\|\mathbb{E}_{\boldsymbol{z}_m}\left[\nabla\mathcal{R}(\widehat{h}_{m-1})-u_m\right]\|\|\widehat{h}_{m-1}-h^\star\|\right]$$

$$\leq D\mathbb{E}_{\boldsymbol{z}_{-m}}\left[\|\mathbb{E}_{\boldsymbol{z}_m}\left[\nabla\mathcal{R}(\widehat{h}_{m-1})-u_m\right]\|\right].$$

Then, applying Lemma A.3 and setting $\tau \triangleq D\kappa$ we get

$$\mathbb{E}_{\boldsymbol{z}_{1:M}}\left[\langle\nabla\mathcal{R}(\widehat{h}_{m-1})-u_m,\widehat{h}_{m-1}-h^\star\rangle\right]$$
$$\leq \tau(\widehat{\Phi})\left(\|\Phi-\widehat{\Phi}\|^2_{L^2(\mathbb{P}_X\otimes\mathbb{P}_Z)}+\|r-\widehat{r}\|^2_{L^2(Z)}+\|\mathcal{P}-\widehat{\mathcal{P}}\|^2_{\text{op}}\right)^{\frac{1}{2}}. \qquad (17)$$

All that is left to do is to apply equations (14), (15), (16) and (17) along with the inequality which defines convexity. Let $\widehat{h} \triangleq \frac{1}{M}\sum_{m=1}^{M}\widehat{h}_{m-1}$ and $\xi \triangleq \rho/2$. Then:

$$\mathbb{E}_{\boldsymbol{z}_{1:M}}\left[\mathcal{R}(\widehat{h})-\mathcal{R}(h^\star)\right]$$

$$\leq \frac{1}{M}\sum_{m=1}^{M}\mathbb{E}_{\boldsymbol{z}_{1:M}}\left[\mathcal{R}(\widehat{h}_m)-\mathcal{R}(h^\star)\right]$$

$$\leq \frac{D^2}{2M\alpha_M}+\xi(\widehat{\Phi},\widehat{r},\widehat{\mathcal{P}})\frac{1}{M}\sum_{m=1}^{M}\alpha_m$$

$$+\tau(\widehat{\Phi})\left(\|\Phi-\widehat{\Phi}\|^2_{L^2(\mathbb{P}_X\otimes\mathbb{P}_Z)}+\|r-\widehat{r}\|^2_{L^2(Z)}+\|\mathcal{P}-\widehat{\mathcal{P}}\|^2_{\text{op}}\right)^{\frac{1}{2}}. \qquad \square$$

# B   Comparison with other methods

In this section, we provide some theoretical comparisons with other notable methods for NPIV estimation.

## B.1   KIV

### B.1.1   Risk bound

In our formulation, we substitute the task for estimating $\mathcal{P}^*$ by that of estimating the ratio of densities $\Phi$. In this way, we bypass the need to evaluate an operator estimator $\widehat{\mathcal{P}^*}$ in every iteration, replacing this by evaluating $\widehat{\Phi}$ once and multiplying it by the derivative of the loss function (cf. Algorithm 1). Aside from the computational advantage this poses, it is interesting to investigate the differences which arise in theoretical bounds when comparing with a method which ends up using an estimator of $\mathcal{P}^*$.

In KIV, the authors propose a first stage procedure which estimates $E$ ($\mathcal{P}$, in our notation) and, in the second stage, they use the adjoint of this operator through the kernel mean embedding $\mu(z)=E^*\phi(z)$ ($\phi$ is the feature map, see the notation in [34]). To estimate $E^*$, they use the adjoint of their estimator $E_\lambda^n$ of $E$, where $n$ is the sample size and $\lambda$ is a regularization parameter.

In our Theorem 4.3, provided the learning rate has been appropriately chosen, the excess risk is upper bounded by $\|\Phi-\widehat{\Phi}\|+\|r-\widehat{r}\|+\|\mathcal{P}-\widehat{\mathcal{P}}\|$. In fact, $\|r-\widehat{r}\|$ is just the risk for a simple regression

problem, and so we will focus our analysis on the other two terms. In KIV, the excess risk is bounded using [37, Theorem 2], according to which it is smaller than the sum of five terms. We reproduce this bound here for a more detailed analysis:

**Theorem B.1** (Proposition 32 of [34]). *The excess risk of KIV's stage 2 estimator $\widehat{H}_\xi^m$ can be bounded by five terms:*

$$\mathcal{E}(\widehat{H}_\xi^m) - \mathcal{E}(H_\rho) \le 5[S_{-1} + S_0 + \mathcal{A}(\xi) + S_1 + S_2], \tag{18}$$

*where*

$$S_{-1} = \|\sqrt{T} \circ (\widehat{\mathbf{T}} + \xi)^{-1}(\widehat{\mathbf{g}} - \mathbf{g})\|_{\mathcal{H}_\Omega}^2,$$
$$S_0 = \|\sqrt{T} \circ (\widehat{\mathbf{T}} + \xi)^{-1} \circ (\mathbf{T} - \widehat{\mathbf{T}})H_\xi^m\|_{\mathcal{H}_\Omega}^2,$$
$$S_1 = \|\sqrt{T} \circ (\mathbf{T} + \xi)^{-1}(\mathbf{g} - \mathbf{T}H_\rho)\|_{\mathcal{H}_\Omega}^2,$$
$$S_2 = \|\sqrt{T} \circ (\mathbf{T} + \xi)^{-1} \circ (T - \mathbf{T})(H_\xi - H_\rho)\|_{\mathcal{H}_\Omega}^2,$$
$$\mathcal{A}(\xi) = \|\sqrt{T}(H_\xi - H_\rho)\|_{\mathcal{H}_\Omega}^2.$$

In our notation, $\mathcal{E}$ corresponds to $\mathcal{R}$, $\widehat{H}_\xi^m$ corresponds to $\widehat{h}$ and $H_\rho$ corresponds to $h^\star$. The space $\mathcal{H}_\Omega$ is a RKHS which contains the solution $H_\rho$. It is also important to recall the definition of the terms in bold, since they depend on $\mu_\lambda^n(z) = (E_\lambda^n)^*\phi(z)$:

$$\mathbf{T} = \frac{1}{m}\sum_{i=1}^m T_{\mu(\tilde{z}_i)}, \quad \mathbf{g} = \frac{1}{m}\sum_{i=1}^m \Omega_{\mu(\tilde{z}_i)}\tilde{y}_i$$
$$\widehat{\mathbf{T}} = \frac{1}{m}\sum_{i=1}^m T_{\mu_\lambda^n(\tilde{z}_i)}, \quad \widehat{\mathbf{g}} = \frac{1}{m}\sum_{i=1}^m \Omega_{\mu_\lambda^n(\tilde{z}_i)}\tilde{y}_i.$$

Where $\tilde{z}_i$ and $\tilde{y}_i$ are second-stage samples. We refer the reader to [34] for a detailed description of the other mathematical objects. Here, we will argue that

$$S_{-1} + S_0 \quad \text{is analogous to} \quad \|\Phi - \widehat{\Phi}\| + \|\mathcal{P} - \widehat{\mathcal{P}}\|. \tag{19}$$

The other terms in Equation (18) are controlled by optimally choosing how the regularizing parameter $\xi$ goes to zero. This relates to the first two terms of our bound in Equation (10) that goes to zero as the SGD iteration increases, as long as we properly choose the learning rate. In [34, Proposition 35], these two terms are bounded as follows:

$$S_{-1} \le \|\sqrt{T} \circ (\widehat{\mathbf{T}} + \xi)^{-1}\|_{\mathcal{L}(\mathcal{H}_\Omega)}^2 \|\widehat{\mathbf{g}} - \mathbf{g}\|_{\mathcal{H}_\Omega}^2,$$
$$S_0 \le \|\sqrt{T} \circ (\widehat{\mathbf{T}} + \xi)^{-1}\|_{\mathcal{L}(\mathcal{H}_\Omega)}^2 \|\widehat{\mathbf{T}} - \mathbf{T}\|_{\mathcal{L}(\mathcal{H}_\Omega)}^2 \|H_\xi^m\|_{\mathcal{H}_\Omega}^2.$$

Then, [34, Proposition 37] refers to [37, Supplements 7.1.1 and 7.1.2] to bound $\|\widehat{\mathbf{g}} - \mathbf{g}\|^2$ and $\|\widehat{\mathbf{T}} - \mathbf{T}\|^2$. In [37], one can see that these bounds amount to[11]

$$\|\widehat{\mathbf{g}} - \mathbf{g}\|^2 \le C \cdot \frac{1}{m}\sum_{i=1}^m \|\mu_\lambda^n(\tilde{z}_i) - \mu(\tilde{z}_i)\|_{\mathcal{H}_\mathcal{X}}^2 |\tilde{y}_i|^2$$

$$\|\widehat{\mathbf{T}} - \mathbf{T}\|^2 \le D \cdot \frac{1}{m}\sum_{i=1}^m \|\mu_\lambda^n(\tilde{z}_i) - \mu(\tilde{z}_i)\|_{\mathcal{H}_\mathcal{X}}^2,$$

where $C$ and $D$ are constants. Then, KIV uses the bound on $\|\mu_\lambda^n(\tilde{z}_i) - \mu(\tilde{z}_i)\|_{\mathcal{H}_\mathcal{X}}^2$ provided in their Corollary 1, which is based on the inequality[12]

$$\|\mu_\lambda^n(\tilde{z}_i) - \mu(\tilde{z}_i)\|_{\mathcal{H}_\mathcal{X}}^2 \le \|E_\lambda^n - E_\rho\|_{\mathcal{H}_\Gamma}^2 \|\phi(\tilde{z}_i)\|_{\mathcal{H}_\mathcal{Z}}^2$$

---

[11]In [37] there is a $2h$ in the exponent instead of 2, which is translated to $2\iota$ in KIV. However, this is only a matter of assuming Holder continuity of $\Omega$ with exponent $\iota$. If one assumes Lipschitz continuity, the exponent becomes 2.

[12]$E_\rho = E$.

Hence, with high probability and ignoring constants, we have[13]

$$\|\widehat{\mathbf{g}} - \mathbf{g}\| \lesssim \|E_\lambda^n - E_\rho\|$$
$$\|\widehat{\mathbf{T}} - \mathbf{T}\| \lesssim \|E_\lambda^n - E_\rho\|.$$

This would be an intermediate result in [34, Proposition 37].

Therefore, the estimation error of $E_\lambda^n$ when compared to $E$ impacts the risk of KIV's estimator through two terms. These are analogous to our two terms $\|\Phi - \widehat{\Phi}\|$ and $\|\mathcal{P} - \widehat{\mathcal{P}}\|$, since $\|\Phi - \widehat{\Phi}\|_{L^2(\mathbb{P}_X \otimes \mathbb{P}_Z)}$ is the Hilbert-Schmidt norm of the difference between $\mathcal{P}^*$ and the integral operator induced by $\widehat{\Phi}$ [8, Theorem 2.42].

### B.1.2 Binary outcomes

In Section 6, we show that, under the binary outcomes setting, the correct formulation of the risk is

$$\mathcal{R}(h) = \mathbb{E}[\mathrm{BCE}(r(Z), F(\mathcal{P}[h](Z)))].$$

In KIV's notation, this would become[14]

$$\mathcal{E}(H) = \mathbb{E}[\mathrm{BCE}(Y, F(H\mu(Z)))]$$

Substituting the populational mean by the empirical one, taking into account that we must estimate $\mu$ and applying regularization, we get (cf. [34, Section 4.2])

$$\widehat{\mathcal{E}}_\xi^m(H) = \frac{1}{m} \sum_{i=1}^m \mathrm{BCE}(\tilde{y}_i, F(H\mu_\lambda^n(\tilde{z}_i))) + \xi\|H\|_{\mathcal{H}_\Omega}^2.$$

Using the fact that $\mathcal{H}_\Omega$ and $\mathcal{H}_\mathcal{X}$ are isometrically isomorphic [34, Section 4.2], we can formulate the risk in an equivalent way, now defined on $\mathcal{H}_\mathcal{X}$:

$$\widehat{\mathcal{E}}_\xi^m(h) = \frac{1}{m} \sum_{i=1}^m \mathrm{BCE}(\tilde{y}_i, F((E_\lambda^n h)(\tilde{z}_i))) + \xi\|h\|_{\mathcal{H}_\mathcal{X}}^2.$$

Then, the KIV estimator is defined as

$$\widehat{h}_\xi^m = \underset{h \in \mathcal{H}_\mathcal{X}}{\arg\min} \ \widehat{\mathcal{E}}_\xi^m(h).$$

For the quadratic loss, KIV finds the solution in closed form, since this is a ridge regression problem. However, for this new loss, no obvious closed form solution exists. Applying the Representer Theorem, we find that the solution $h_\xi^m$ should satisfy

$$h_\xi^m = \sum_{i=1}^n \alpha_i \psi(x_i),$$

for some $\alpha \in \mathbf{R}^n$, where $\psi(x_i) = K(x_i, \cdot)$ is the reproducing kernel of $\mathcal{H}_\mathcal{X}$. Then, one is left with the following optimization problem:

$$\underset{\alpha \in \mathbf{R}^n}{\arg\min} \ \frac{1}{m} \sum_{i=1}^m \mathrm{BCE}(\tilde{y}_i, F(\alpha' w(\tilde{z}_i)) + \xi\alpha' K_{XX}\alpha$$

Since $(E_\lambda^n h_\xi^m)(z) = \alpha' w(z)$ (see the definition of $w$ in [34, Appendix A.5.1]). An approximate solution to this $n$-dimensional optimization problem must be found with numerical methods. Keep in mind that this problem must be solved *several times* in order to find a good value of the regularization parameter $\xi$ via cross-validation, which is crucial for the performance of regularized regression methods. In contrast, our method does not require additional work after defining the modified risk.

---

[13]We use the symbol $\lesssim$ to mean that the LHS is smaller or equal than a constant times the RHS, where the constant does not depend on data.

[14]Since BCE is linear in the first argument, it does not matter if we use $Y$ or $r(Z)$ to define the risk.

## B.2    Dual IV

In DualIV, the risk minimization problem is reformulated into a saddle point optimization problem, which we write in their notation:

$$\min_{f \in \mathcal{F}} \max_{u \in \mathcal{U}} \Psi(f, u),$$

where

$$\Psi(f, u) = \mathbb{E}_{XYZ}[f(X)u(Y, Z)] - \mathbb{E}_{YZ}[\ell_Y^\star(u(Y, Z))].$$

Here, $\mathcal{F}$ is the space which contains the solution to the NPIV estimation problem, $\mathcal{U}$ is the space of the "dual function" and $\ell_y^\star$ is the Fenchel dual of the function $\ell_y = \ell(y, \cdot)$. In the binary outcomes setting, the loss is given by $\ell(y, y') = \mathrm{BCE}(y, F(y'))$. If $F$ is the logistic function, i.e. $F(y') = \exp(y')/(1 + \exp(y'))$, one may verify that

$$\ell_y^\star(u) = \begin{cases} -H(u + y), & \text{if } 0 \leq u + y \leq 1, \\ +\infty, & \text{otherwise,} \end{cases}$$

where $H(p) = -[p \log p + (1 - p) \log(1 - p)]$ is the entropy. The empirical minimax objective then becomes

$$\Psi(f, u) = \frac{1}{n} \sum_{i=1}^{n} f(x_i)u(y_i, z_i) + \frac{1}{n} \sum_{i=1}^{n} H(y_i + u(y_i, z_i)),$$

with the constrain that $0 \leq y_i + u(y_i, z_i) \leq 1$ for every $i$. Again, in contrast with the case of quadratic loss, when one may choose $\mathcal{F}$ and $\mathcal{U}$ to be RKHS and obtain a closed form ridge-regularized solution, this formulation is not as tractable.

## B.3    GMM

In GMM based methods for continuous outcomes, the main idea is that since $\mathbb{E}[\varepsilon \mid Z] = 0$ and $Y = h^\star(X) + \varepsilon$, we have $\mathbb{E}[Y - h^\star(X) \mid Z] = 0$ and, hence, $\mathbb{E}[f(Z)(Y - h^\star(X))] = 0$ for every $f \in L^2(Z)$. If we choose a large number of functions $f_1, \ldots, f_m$, we may search for $h^\star$ by finding a function which minimizes

$$\sum_{j=1}^{m} \psi_n(h, f_j)^2 \quad \text{where} \quad \psi_n(h, f_j) = \frac{1}{n} \sum_{i=1}^{n} f_j(Z_i)(Y_i - h(X_i)). \tag{20}$$

This is possible because the error term $\varepsilon$ can be written as $Y - h^\star(X)$. Under the structural equation $Y = \mathbf{1}\{h^\star(X) + \varepsilon > 0\}$, such a characterization of the error term is not available, which already presents a difficulty. Nonetheless, as we show in Section 6, under this new equation we have $\mathbb{E}[Y \mid Z] = F(\mathcal{P}[h^\star](Z))$. Therefore, moment conditions could be obtained through $\mathbb{E}[(Y - F(\mathcal{P}[h^\star](Z)))f(Z)] = 0$ for every $f \in L^2(Z)$. However, this would create the need to introduce an estimator $\widehat{\mathcal{P}}$ of $\mathcal{P}$. This defeats one of the main advantages of turning to GMM, which is to avoid the first stage of two-stage methods, since errors in the first stage can have a large impact on the resulting estimator.

## C    Implementation and experiment details

In this section, we give practical guidelines for implementing SAGD-IV, as well further details regarding the baseline methods below:

- KIV: Since the original implementation was only publicly available in Matlab, to facilitate reproducibility of our results we re-implemented this method in Python. This was done following the guidelines presented in the paper.

- DualIV: Similarly to KIV, we rewrote the method in Python using the closed form solutions and the hyperparameter selection procedure present in the paper.

- DeepGMM: We used the publicly available Python implementation[15].

- DeepIV: We used the implementation available as part of the EconML package [5].

---

[15] https://github.com/CausalML/DeepGMM.git

- 2SLS: We include the classic Two Stage Least Squares procedure to show the benefits of following a nonparametric approach.

All experiments were conducted on a Apple Sillicon M1 Pro CPU. No GPUs were used.

*Remark* C.1 (Experiments sample size). All four baseline methods we are considering need a single dataset $(X, Z, Y)$ to fit their estimator (hyperpameter validation included), whereas our method needs one dataset of $(X, Z, Y)$ triplets to compute $\widehat{\Phi}, \widehat{r}$ and $\widehat{r}$, and one dataset of $Z$ samples to conduct the SAGD loop. To make fair comparisons, we trained each method using the same **total amount of random variable samples**. For example, DeepGMM might use 800 triplets $(X, Z, Y)$ for training and 200 triplets for validation, whereas SAGD-IV may use 600 triplets for fitting the preliminary estimators and then 1200 samples of $Z$ for the SAGD loop, both totaling 3000 *random variable samples* used to adjust the model.

## C.1 SAGD-IV

### C.1.1 On computing $\widehat{\mathcal{P}}, \widehat{\Phi}$ and $\widehat{r}$

**Ratio of Densities.** Density ratio estimation is a well studied problem in Machine Learning, with [36] being a good resource for acquainting oneself with relevant algorithms. The approaches we follow here consist on obtaining an estimator $\widehat{\Phi}$ that minimizes the (empirical version of the) quantity $\frac{1}{2}\|\widehat{\Phi} - \Phi\|^2_{L^2(\mathbb{P}_X \otimes \mathbb{P}_Z)}$. We consider two options for $\widehat{\Phi}$: a Reproducing Kernel Hilbert Space (RKHS) estimator obtained through the Unconstrained Least Squares Importance Fitting (uLSIF) framework described in [36] and also a neural network trained to minimize the same objective over its parameters. By changing the kernel or the activation function used, we can make sure this estimator adheres to Assumption 4.2 Item 4.

**Conditional expectation operator.** This is the most complex object we must estimate, since it encompasses all possible regressions of functions of $X$ over $Z$. Not many options exist for this task, with a popular one [13, 8] being the use of Nadaraya-Watson kernels [26, 39]. We, however, chose to employ the estimation method used as a first stage in KIV [34], which transforms the problem into vector-valued RKHS ridge regression, due to its straightforward implementation.

**Conditional mean of $Y$ given $Z$.** This, in contrast, is the simplest unknown in our formulation, being the regression function of $Y$ over $Z$. We again consider two options for this task: the first is to employ the method previously discussed in order to estimate the conditional expectation operator of $Y$ given $Z$, and then apply it to the identity function. The second option is to train a general purpose regressor, such as a neural network.

### C.1.2 Hyperparameters

Since the final estimator in Algorithm 1 is an average over the whole path of estimates produced by the SAGD loop, we can improve upon its quality by discarding the first $K$ estimates $\widehat{h}_1, \ldots, \widehat{h}_K$, where $K$ must be chosen independently from $M$. In this way, it is as if we started our algorithm from $\widehat{h}_{K+1}$, after a warm up period.

Regarding sample splitting, our theoretical results do not specify an optimal ratio between the $M$ SAGD iterations and the $N$ samples of $(X, Z, Y)$ used to compute $\widehat{\Phi}, \widehat{r}$ and $\widehat{\mathcal{P}}$. We empirically determined that $M = 2N$ is a good guideline to follow, and it is the ratio we employed in Section 5. Of course, in real data scenarios this ratio might be constrained due to limited data availability.

About hyperparameters:

- The Algorithm 1 learning rate was set to be $\alpha_m = 1/\sqrt{M}$ for $1 \le m \le M$;
- The warm up time $K$ was set to 100;
- The set $\mathcal{H}$ was chosen as in Equation (6), with $A$ set to 10. During our experiments, we found that this parameter, aside from needing to be large enough for us to have $h^\star \in \mathcal{H}$, did not have any noticeable influence on the resulting estimator.
- For **Kernel SAGD-IV**:

- We used Gaussian kernels, with lenghtscale parameter determined through the median heuristic;

- The regularization parameters for the three kernel ridge regressions were chosen through cross validation. Since these parameters are essential for actual learning to take place, we selected the values to be tested through an iterative procedure, described in Algorithm 2.

- For **Deep SAGD-IV**:

  - For both networks in the roles of $\widehat{\Phi}$ and $\widehat{r}$, we used two hidden dense layers of sizes $64$ and $32$. In the continuous response setting the activation function used was a ReLU, while for the binary response one we used the sigmoid activation. For $\widehat{\Phi}$, we used dropout with rate $0.01$.

  - Both networks were trained with a batch size of $512$, using the Adam [21] optimizer with learning rate of $0.01$, accross $1.5 \cdot 10^5 / N$ epochs. We adopted $L^2$ regularization in the two estimators, with regularization parameters of $3 \cdot 10^{-3}$ for $\widehat{r}$ and $5 \cdot 10^{-3}$ for $\widehat{\Phi}$. We also implemented early stopping in the two training procedures.

- We remind the reader that both SAGD-IV variants use kernel methods to compute $\widehat{\mathcal{P}}$.

## C.2 DeepGMM

The modifications we made with respect to the original paper were the following:

- Instead of using $50\%$ of the samples for training and $50\%$ for validation, we adopted a train / validation split of 80 / 20.

- We changed the batch size from $1024$ to $256$. This was done because we supplied the algorithm with $1000$ training samples instead of the $2000$ used in [6], and hence a batch size of $1024$ would not produce stochastic gradients.

## C.3 DeepIV

We followed all the guidelines presented in the original paper for low dimensional settings.

## C.4 KIV

We employed a $50/50$ split between $(X, Z, Y)$ and $(\tilde{X}, \tilde{Z}, \tilde{Y})$ observations (see [34], Appendix A.5.2). Regularization parameter candidates were also chosen using the selection method in Algorithm 2.

---

**Algorithm 2** Search for regularization parameter

---

**Input:** Loss function $L : \mathbf{R} \to \mathbf{R}$. List of initial values **parameters** $= [\lambda_1, \ldots, \lambda_n]$. Initial offset $\varepsilon$. Number of iterations $T$.
**Output:** $\lambda^\star$
**for** $1 \leq t \leq T$ **do**
   $\lambda^\star \leftarrow \arg\min \{L(\lambda) \text{ for } \lambda \text{ in } \textbf{parameters}\}$
   **parameters** $\leftarrow [\lambda^\star + k\varepsilon \text{ for } k \in \{-5, -4, \ldots, 5\}]$
   $\varepsilon \leftarrow \varepsilon/10$
**end for**

---

## C.5 DualIV

We followed the hyperparameter selection guidelines given in Section 4 of [25]. This means we used a $50/50$ split for train/validation samples and chose the regularization parameters from a grid search, minimizing the squared norm of the dual function $u$.

# D  Small data regime

This section presents a version of the experiments in Section 5.1 with half the sample size, providing a comparison between SAGD-IV and other methods within a smaller data availability regime. Figure 3 contains the corresponding log-MSE results.

Contrasting Figures 1 and 3, we can see that the relative performances for each method mostly stay the same. Some methods, such as Deep-IV in the **abs** scenario, exhibited an error distribution with larger variance, but this is to be expected from the reduction in sample size, which makes any given dataset realization a worse depiction of the true data distribution. These results support the conclusion that our method's performance is competitive with other state-of-the-art approaches in more data availability conditions.

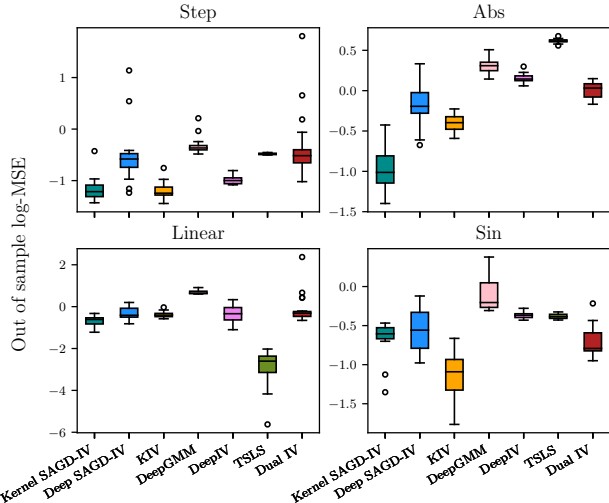

Figure 3: Log-MSE results for the experiments in Section 5.1 with half the sample size.

# E  Alternative risk definition

Computing the risk using $r(Z)$ or $Y$ are equivalent under the quadratic loss, however, one might think that considering $Y$ instead of $r(Z)$ in our formulation would lead to an algorithm that does not need to estimate $r(Z)$. Below we show that this is not the case, since the gradients of the risk measure under both formulations are the same. This implies that we would still need to estimate $r$ for the algorithm.

If instead of Equation (5), we consider the risk defined as

$$\widetilde{\mathcal{R}}(h) = \mathbb{E}[\ell(Y, \mathcal{P}[h](Z))],$$

then proceeding in exactly the same way as in the proof of Proposition 3.3, we would arrive at

$$D\widetilde{\mathcal{R}}[h](f) = \mathbb{E}[\partial_2\ell(Y, \mathcal{P}[h](Z)) \cdot \mathcal{P}[f](Z)]$$

As it is, this expression is not an inner product in $L^2(Z)$ since $Y$ is not measurable with respect to $Z$. However, if $\partial_2\ell$ is linear with respect to its first argument, which is true for the quadratic and binary cross entropy losses, we have

$$
\begin{aligned}
D\widetilde{\mathcal{R}}[h](f) &= \mathbb{E}[\partial_2\ell(Y, \mathcal{P}[h](Z)) \cdot \mathcal{P}[f](Z)] \\
&= \mathbb{E}\left[\mathbb{E}[\partial_2\ell(Y, \mathcal{P}[h](Z)) \cdot \mathcal{P}[f](Z) \mid Z]\right] \\
&= \mathbb{E}\left[\partial_2\ell(\mathbb{E}[Y \mid Z], \mathcal{P}[h](Z)) \cdot \mathcal{P}[f](Z)\right] \\
&= \mathbb{E}\left[\partial_2\ell(r(Z), \mathcal{P}[h](Z)) \cdot \mathcal{P}[f](Z)\right],
\end{aligned}
$$

which is the same expression obtained before. Hence, the gradient of $\mathcal{R}$ still depends on $r(Z)$ and the resulting method still needs to estimate this quantity.

Nonetheless, one could still use the sample $y_i$ as an estimate of $r(z_i)$. However, this is not the best choice, because the term $\|\widehat{r} - r\|^2_{L^2(Z)}$ which appears in the RHS of Equation (10) would become $\mathbb{E}[\mathrm{Var}[Y \mid Z]]$. This is constant with respect to the size of the auxiliary dataset $\mathcal{D}$ (cf. Assumption 4.2) and could be improved by taking $\widehat{r}$ to be a better regressor of $Y$ over $Z$.

