# OpenReview forum: "Nonparametric Instrumental Variable Regression through Stochastic Approximate Gradients"
_NeurIPS.cc/2024/Conference — NeurIPS 2024 poster_

### Official Review · Reviewer_xcFr · 2024-06-25

**Soundness:** 3
**Presentation:** 3
**Contribution:** 2
**Rating:** 5
**Confidence:** 5

**Summary:**

This paper studies the nonparametric instrumental variable (NPIV) regression problem. The authors present a new algorithm, SAGD-IV, which utilizes stochastic gradient descent in a function space to directly minimize the populational risk. The gradient can be computed via computing the estimating the conditional density $\Phi$ and the conditional expectation operator $\mathcal{P}$. This approach is distinguished by its flexibility, accommodating a variety of supervised learning algorithms, such as neural networks and kernel methods, and supporting both continuous and binary outcome scenarios. Finally, the authors also proved the consistency of the NPIV estimator under regularity assumptions.

**Strengths:**

Estimating the gradient to perform SGD in the second stage for NPIV appears to be novel.
The algorithms can also be applied to non-quadratic loss function is also new.

**Weaknesses:**

I have several concerns of the paper:

1 The author claims that existing methods in estimating NPIV solutions face the problem of cannot apply to large, high dimensional datasets. Typically in the NPIV context, such a problem arises due to the estimation of the condtional density function $\Phi$. However, in the paper, computing the gradient also requires us to compute the conditonal density fucntion, so I do not see how such limitation on high dimensional dataset can be avoided.

2. The novelty of the algorithm relies on performing the sgd in the second stage after we estimate $\Phi$ and $\mathcal{P}$. I believe in standard two stage estimation methods, once we obtain $\mathcal{P}$ (no estimationf of $\Phi$ required), the conditional expectation operator, we could also apply standard gradient descent to obtain the final estimator. I do not see how the proposed algorithm is significantly different from the standard ones. However, the proposed algorithm have the additional requirement to compute the conditional density function which is significantly harder than computing $\mathcal{P}$.

3. The conducted experiments is not convicing as even in the low dimensional regime, the obtained mse is not significantly better than the standard two stage estimation method. And the experiments do not contain the high dimensional setting. So it is not clear how the method works in such settings.  A nolvety the authors seem to claim.

**Questions:**

See weakness.

---

> ### Author Rebuttal · Authors · 2024-08-06
>
> We thank the reviewer for the insightful observations. Below, we separately address each weakness:
>
> 1. The referee is correct in pointing out that estimating conditional densities in high dimensional settings poses nontrivial difficulties, being a limitation which applies in our case, as well as in other approaches to the NPIV problem. In Section 1.1, we start by noting that classical NPIV methods, like those based on sieves or Nadaraya-Watson kernel estimators, exhibit difficulties when dealing with large, high dimensional datasets, while more modern approaches leverage deep learning techniques in an attempt to overcome these issues, but end up susceptible to higher prediction variance under the opposing scenario of few data points. Our goal with this discussion is to point out that, since our formulation is agnostic with respect to how the density ratio is estimated, it could be tailored to specific scenarios and leverage recent advances from other areas. For example, one could use neural networks in large data/dimension scenarios, and kernel methods for situations where there is not a lot of data/the data is low dimensional.
>
> 2. We would argue that estimating $\Phi$ is not significantly harder than estimating $\mathcal{P}$, as the latter is an operator acting on an infinite dimensional space. This is evidenced by the very few results discussing convergence rates for estimators of this conditional expectation operator. Furthermore, according to Proposition 3.3., in our loss-agnostic framework, the gradient at $h$ is given by the application of the adjoint operator $\mathcal{P}^*$ at the composition of $\partial_2 \ell$ with $\mathcal{P}[h]$. So, any gradient-based algorithm needs to somehow tackle the estimation of both $\mathcal{P}$ and $\mathcal{P}^*$ as operators. For instance, in [1] they consider sieve estimators using separate basis expansions for $\mathcal{P}$ and $\mathcal{P}^*$. Our stochastic gradient formulation is able to overcome this problem by substituting the outer application of $\mathcal{P}^*$ with a simple multiplication by $\Phi$.
>
> 3. To our understanding, in Figure 1, our MSE is lower and statistically better than the standard two-stage method, except for the case in which TSLS is correctly specified. Note, however, that this is expected against any method, as it falls within the realm of parametric estimation. The log-MSE shows significant improvement over the other methods. In particular, the Deep learning based methods (DeepGMM, DeepIV) show larger, more variable log-MSE than the two variants of SAGD-IV. The same is true for Dual IV. The strongest competitor is the KIV method, which demonstrates similar log-MSE for the step and linear function scenarios, lower log-MSE for sine (but with larger variance) and statistically higher log-MSE in the Abs scenario. Therefore, we argue that the numerical experiment indicates that our method is at least comparable with the state-of-the-art methodologies.
>
>    As indicated in our answer to the reviewers first point, we are not claiming that our method is the first to address the high-dimensional setting, we have just pointed out the flexibility of our framework to possibly deal with such scenarios. Given the short reviewing period and the computational resources available to us, unfortunately, we could not run the experiments again in high-dimensional settings.
>
> **References**
>
> [1] Darolles, S., Fan, Y., Florens, J. P., & Renault, E. (2011). NONPARAMETRIC INSTRUMENTAL REGRESSION. Econometrica, 79(5), 1541–1565. http://www.jstor.org/stable/41237784

---

> > ### Comment · Reviewer_xcFr · 2024-08-08
> >
> > Thanks for the reply. I will maintain my score.

---

### Official Review · Reviewer_spVa · 2024-07-11

**Soundness:** 4
**Presentation:** 4
**Contribution:** 3
**Rating:** 8
**Confidence:** 5

**Summary:**

The authors propose an estimator for nonparametric instrumental variable regression (NPIV), with an extension to the binary outcome case. They prove that the excess risk of the projected estimator is controlled by rates of the density ratio, a regression, and the conditional expectation operator. Extensive comparative simulations are conducted. Careful analysis in the appendix also compares this approach with others.

**Strengths:**

The main result is impressively agnostic about the component estimators.

The paper is very well written, and I particularly appreciated the authors’ comparison across formulations and notations in the literature.

**Weaknesses:**

Algorithm 1 of ''Provably Efficient Neural Estimation of Structural Equation Model: An Adversarial Approach’’ by Liao et al. (2020) is another stochastic approximate gradient approach for NPIV. I believe the difference in the algorithms amounts to whether the primal or dual formulation is optimized with stochastic gradients. A clarification would be welcome.

It would be good to put into words that the excess risk being analyzed is for objects projected onto the instrument space, which is common in this literature but perhaps less so in the NeurIPS community.

**Questions:**

Are there known rates for the density ratio given in the first term of zeta? Please provide at least one reference where a rate in this norm, or a sufficient norm, is given.

Similarly, it would be good to point to rates for the third term of zeta, such as Theorem 2 of ''Kernel Instrumental Variable Regression'' by Singh et al. (2019), Theorem 5 of ''Sobolev Norm Learning Rates for Conditional Mean Embeddings'' by Talwai et al. (2022), and Proposition S5 of ''Kernel Methods for Causal Functions: Dose, Heterogeneous and Incremental Response Curves'' by Singh et al. (2024).

---

> ### Author Rebuttal · Authors · 2024-08-06
>
> We thank the reviewer for the attentive analysis and relevant suggestions. Below, we separately address the weaknesses and questions.
>
> ### **Weaknesses**
>
> Thank you very much for pointing out such an interesting paper. The cited work explicitly uses NN classes as approximations to $L^2$ spaces and conducts SGD on the *weights* of these neural nets. This is different from our approach, which formulates the SGD updates *within the function space $L^2(X)$* and does not assume a parametric form for the stochastic gradients. Another difference is that our work directly addresses the primal problem and Liao et al. (2020) converts it to a saddle-point problem in order to apply a primal-dual algorithm, as you mentioned. Additionally, we do not have any regularization and we do not assume $r$ is known (page 5, paragraph 2, $b$ in their notation). We will include this discussion in the literature review.
>
> Regarding the matter of "risk" versus "projected risk", we wholly agree and will include an appropriate remark in Section 3.
>
> ### **Questions**
>
> In [1, Theorem 14.16] the authors provide a convergence rate for our proposed norm of $((\log n) / n)^{1/(2 + \gamma)}$, where $\gamma$ controls the decay of the spectrum of the kernel used to estimate the density ratio.
>
> Concerning the references for the operator regression convergence rate, we thank the reviewer for bringing the second and third to our attention, which will be included in the final version of the paper.
> We could not find Proposition S5 in the third paper but if the referee can clarify this misunderstanding, we would appreciate it.
> We will add a paragraph in Subsection 4.2. pointing out these known rates.
>
> **References**
>
> [1] Sugiyama M, Suzuki T, Kanamori T. Density Ratio Estimation in Machine Learning. Cambridge University Press; 2012.

---

> > ### Comment · Reviewer_spVa · 2024-08-09
> >
> > Thank you for these replies.
> >
> > Proposition S5 in the third paper can be found in its Online Supplement on the Biometrika website.

---

> > > ### Author Response · Authors · 2024-08-09
> > >
> > > Thanks for the clarification, we will add the reference in the final version of the paper.

---

### Official Review · Reviewer_1tRc · 2024-07-12

**Soundness:** 3
**Presentation:** 3
**Contribution:** 2
**Rating:** 5
**Confidence:** 3

**Summary:**

This paper considers the standard problem of non-parametric instrumental variable estimation (NPIV) and proposes a new approach of functional stochastic gradient descent to solve it (SAGD–IV), where the gradient estimator can be implemented and adapted using certain machine learning or deep learning techniques. Theoretical guarantee of the finite-time convergence of the algorithm is provided under suitable assumptions. Simulated experiments show the effectiveness of the proposed method to some extent.

**Strengths:**

**Originality and Significance:**

1. The idea of using functional gradient descent to solve the NPIV problem is somehow new and interesting, which is different from most previous relevant works.
2. The derivation of the functional gradient form in Equation (8) is important since it allows us to decouple the estimation of $\mathcal{P}$ and $\mathcal{P}^{\ast}$ and gives a computationally efficient way to estimate the gradient of the risk functional.
3. The algorithm can be easily adapted to different machine learning techniques to estimate the functional gradient, and can also be adapted to either continuous or binary outcomes as demonstrated in the experiments.

**Clarity and Quality:**

The presentation is clear. The problem setup is nicely formulated and all the assumptions required are well stated. The main theoretical result is also sound and is well proved.

**Weaknesses:**

1. The consistency or sample-complexity of the proposed method is unknown. This type of finite sample results are presented in recent previous methods on NPIV including but not limited to [1, 2, 3].

2. The technique and idea behind the proof of the convergence of SAGD-IV is relatively standard and straightforward given that the risk function $\mathcal{R}$ is convex in the functional space.


**References:**

[1] Singh, R., Sahani, M., & Gretton, A. (2019). Kernel instrumental variable regression. Advances in Neural Information Processing Systems, 32.

[2] Bennett, A., Kallus, N., Mao, X., Newey, W., Syrgkanis, V., & Uehara, M. (2023, July). Minimax Instrumental Variable Regression and $ L_2 $ Convergence Guarantees without Identification or Closedness. In The Thirty Sixth Annual Conference on Learning Theory (pp. 2291-2318). PMLR.

[3] Li, Z., Lan, H., Syrgkanis, V., Wang, M., & Uehara, M. (2024). Regularized DeepIV with Model Selection. arXiv preprint arXiv:2403.04236.

**Questions:**

1. Could you please highlight the technical difficulty and the novelty in proving the main theoretical results of the convergence of SAGD-IV?

2. It is not clearly stated how to exactly perform the functional gradient descent step as proposed in Algorithm 1 in the practical experiments. Could the authors elaborate on this a bit more?

3. What is the performance of the proposed algorithm in the small data regime when compared with previous methods?

**Limitations:**

Please see the above weakness section and the question section.

---

> ### Author Rebuttal · Authors · 2024-08-06
>
> We thank the reviewer for providing a careful assessment of our work. In what follows, we address the weaknesses and questions in a point by point fashion.
>
> ### **Weaknesses**
>
> 1. We are assuming that "consistency" here means convergence of $||\hat{h} - h^*||$ to $0$ as the number of data points grows to infinity. (In [1] the authors use this word to mean convergence of the *projected risk* to zero, the same type of convergence we provide in Theorem 4.3.).
>
>    Consistency for NPIV regression is, as we mention in Remark 4.5, a very challenging problem. The cited methods [2] and [3], as well as others which prove this kind of result (e.g. [5]), are based on Ridge regression and, hence,  can leverage the source condition, ubiquitous in the inverse problems literature, to obtain consistency. In the context of SGD, convergence to the global minimum could be achieved when the function being minimized is strongly convex, which, in our context, translates to restrictions on the operator $\mathcal{P}$, as we point out in Remark 4.5. A similar assumption was also made in [4, Theorem 7, Appendix E.4].
>
>    With respect to sample-complexity, note that $|\mathcal{D}|$ (cf. Assumption 4.2) can be seen as the *first stage sample size*, while the number $M$ in Algorithm 1 is the *second stage sample size*. The first two terms on the RHS of Equation (10) quantify the rate of convergence with respect to $M$, while the dependence on $|\mathcal{D}|$ is in the $\zeta$ term. Note that this term is comprised of three estimation errors, whose exact dependence on $|\mathcal{D}|$ can only be known after deciding on specific estimation methods. We did not expand this term further since we chose to be agnostic with respect to the first stage estimators. Nonetheless, if one wants to obtain explicit bounds in $|\mathcal{D}|$, rates for $||\hat{\Phi} - \Phi||$ can be found in [6, Theorem 14.16], while rates for $||\hat{\mathcal{P}} - \mathcal{P}||$ are present in [1, Theorem 2] and [7, Theorem 5]. The other term, $||\hat{r} - r||$, is the risk for a simple real-valued regression problem, and various rates are available depending on the chosen regressor and the degree of smoothness assumed on $r$. The book [8] contains several results of this type.
>
>    We agree that this is a discussion which should be in the main text, and we will update Section 4.2 to include it.
>
> 2. The novelty of our work does not come necessarily from the proof technique of Theorem 4.3, but rather from the insight that, by tackling NPIV with functional stochastic gradient descent and by carefully exploiting its structure, we are able to obtain an algorithm with improved flexibility that is also capable of addressing other loss functions. This reframing is what allows not only the development of an intuitive and competitive algorithm for the challenging NPIV problem, but also the use of standard tools from convex analysis to theoretically analyze it, which we believe to be an important contribution to the literature and to the NeurIPS community.
>
>    Moreover, while we agree with the referee that the proof technique is standard, we point out that Theorem 4.3 differs from straightforward convex analysis in the sense that the operator associated with the NPIV inverse problem is unknown. This is addressed in Lemma A.3, where we must carefully analyze a term which usually vanishes in simpler instances of statistical inverse problems.
>
>
> ### **Questions**
>
> 1. See weakness 2.
>
> 2. Thank you for pointing out that more details are needed to parse the algorithm implemention. We will enrich the discussion in Appendix C and the code will be made available. SAGD-IV needs one dataset $\mathcal{D}$ of samples from the triplet $(X, Z, Y)$ to compute $\hat{\Phi}, \hat{\mathcal{P}}$ and $\hat{r}$ (cf. Assumption 4.2 and Remark C.1). Then, it needs another dataset, say, $\mathcal{D}\_Z$, of samples from only $Z$ to conduct the functional gradient descent in Algorithm 1. After one has the estimators $\hat{\Phi}, \hat{\mathcal{P}}$ and $\hat{r}$ in hands, to evaluate $\hat{h}(x)$ one must loop through the samples in $\mathcal{D}\_Z$, computing the stochastic gradient evaluated at $x$ and using it to perform the projected gradient descent step. There are optimizations which can speed up this process, for instance, the term $\partial_{ 2 } \ell \left( \hat{ r } ( z_{ m } ), \hat{ \mathcal{P} } [ \hat{ h }\_{ m - 1 } ] ( z\_{ m } ) \right)$ does not depend on $x$ and, hence, may be left pre-computed. Then, the only effort in computing $\hat{h}(x)$ is in computing $\hat{\Phi}(x, z_m)$ for $1 \leq m \leq M$.
>
> 3. We believe that our experiments already fall within a relatively small data regime. Nonetheless, we agree that it is interesting to explore this direction and we ran the experiments with half the sample size. The results can be found in the attached PDF and will be included in the Appendix. We can see that the relative performance of each method largely stays the same. However, as is common with fewer data points, the log-MSE distribution for each method generally exhibited higher variance.
>
> **References**
>
> [4]  Muandet, K., Mehrjou, A., Lee, S. K., & Raj, A. (2020). Dual instrumental variable regression. Advances in Neural Information Processing Systems, 33, 2710-2721.
>
> [5] Darolles, S., Fan, Y., Florens, J. P., & Renault, E. (2011). NONPARAMETRIC INSTRUMENTAL REGRESSION. Econometrica, 79(5), 1541–1565. http://www.jstor.org/stable/41237784
>
> [6] Sugiyama M, Suzuki T, Kanamori T. Density Ratio Estimation in Machine Learning. Cambridge University Press; 2012.
>
> [7] Talwai, P., Shameli, A., & Simchi-Levi, D. (2022, May). Sobolev norm learning rates for conditional mean embeddings. In International conference on artificial intelligence and statistics(pp. 10422-10447). PMLR.

---

> > ### Comment · Reviewer_1tRc · 2024-08-12
> >
> > Thank you very much for your detailed response to all my questions and concerns! Also I appreciate the additional experiments conducted in the more small data regimes. Given the theoretical contributions and the potentially insights this work could bring to the community, we would update my score to 5.

---

### Official Review · Reviewer_WKaj · 2024-07-12

**Soundness:** 4
**Presentation:** 3
**Contribution:** 4
**Rating:** 8
**Confidence:** 4

**Summary:**

This paper introduces a novel IV method, called the SAGD-IV, which is more efficient and stable in the NPIV regression. Two different variants of the SAGD-IV are given, and a range of comparisons between these variants with the existing methods are given. Moreover, the performance of SAGD is not only shown in a continuous response case but also is extended to a discrete case.

**Strengths:**

The theoretical results are solid, and it also considered different aspects of the limitation, such as the risk while computing the gradient, different types of $h^{*}$ and variants of SAGD-IV for different circumstances.

I really enjoy looking at the comparison between SAGD-IV and existing methods with different options of $h^{*}$.

**Weaknesses:**

There are a lot of mathematical equations and deductions, like theories, assumptions etc, which is great and indicates that you have a pretty solid work, but it is also great to spend a bit more room for storytelling, which will bring people straight to the main point and get your amazing work.

**Questions:**

I wonder if the error term is not additive and but is a non-linear function of $X$ and $\epsilon$, would the SAGD-IV methods still work?

---

> ### Author Rebuttal · Authors · 2024-08-06
>
> We thank the reviewer for the constructive comments!
>
> Concerning the weakness, we appreciate the reviewer's input on our exposition and we will improve the discussion on IV and the comparison with current approaches.
>
> Regarding your question, it is possible to use SAGD-IV for other models. However, it depends on finding an appropriate risk functional $\mathcal{R}$, which itself depends on finding an appropriate pointwise loss $\ell$. With the pointwise loss in hands, our method is immediately applicable after computing the derivative $\partial_2 \ell$.
>
> For a general approach, if we assume that $Y = \Lambda(h^*(X), \varepsilon)$ for some nonlinear function $\Lambda$, we have to find a function $F$ such that $\mathbb{E}[ Y \mid Z ] = \mathbb{E}[\Lambda(h^*(X), \varepsilon) \mid Z ] = F( \mathbb{E}[h^*(X) \mid Z] )$. Then, if $Y$ is real-valued, a possible choice for $\ell$ is the $L^2$ loss $\ell(y,y') = \frac{1}{2} (y - F(y'))^2$, and, if $Y$ is discrete, one may use the binary cross entropy loss $\ell(y, y') = BCE(y, F(y'))$, as was done in Section 6 for the binary outcome setting, where $\Lambda(h(X), \varepsilon) = 1 \\{ h(X) + \varepsilon > 0 \\}$.

---

> > ### Comment · Reviewer_WKaj · 2024-08-09
> > **Response to rebuttal**
> >
> > Thanks for those replies! I will maintain my score.

---

### Author Rebuttal · Authors · 2024-08-06

We thank all reviewers for reading our paper and providing valuable feedback which will certainly result in improvements to the final version.
We have uploaded a PDF containing experiment results which address points raised by reviewer 1tRc.

---

### Author Response · Authors · 2024-08-13
**End of rebuttal period**

As the rebuttal period comes to an end and all reviewers have replied, we would like to thank everyone for taking the time to read and respond to our rebuttals. We are specially grateful for the comments which regard our work as new, interesting, and relevant to the NeurIPS community.

The feedback given by the reviewers has motivated a series of improvements to our paper, including:

* A more detailed discussion on the decay rate for the $\zeta$ term in Theorem 4.3, with the addition of new references for bounding the estimation errors of $\Phi$, $\mathcal{P}$ and $r$;

* More details on the algorithm's implementation and another experiment in a smaller data size setting;

* An improvement on the discussion of IV regression and related works in the introduction.

---

### Decision · Program_Chairs · 2024-09-25

**Decision:**

Accept (poster)

**Comment:**

The majority of reviewers tend to agree that the paper provides a solid contribution to the research area of Nonparametric Instrumental Variable (NPIV) Regression. It proposes to solve NPIS using a functional stochastic gradient descent algorithm, which is novel. The theoretical results are also very thorough. I believe that people working in this area will appreciate this work.